# DETERMINISTIC DIFFUSION FOR SEQUENTIAL TASKS

## ABSTRACT

Diffusion models have been used effectively in sequential tasks such as video prediction and robot trajectory generation. Existing approaches typically predict sequence segments autoregressively by denoising Gaussian noise. This iterative denoising process is time-consuming, a problem compounded by the autoregressive nature of sequence prediction. In this paper, we aim to expedite inference by leveraging the properties of the sequence prediction task. Drawing on recent work on deterministic denoising diffusion, we initialize the denoising process with a non-Gaussian source distribution obtained using the context available when predicting sequence elements. Our main insight is that starting from a distribution more closely resembling the target enables inference with fewer iterations, leading to quicker generation. We demonstrate the effectiveness of our method on diffusion for video prediction and in robot control using diffusion policies. Our method attains faster sequence generation with minimal loss of prediction quality, in some cases even improving performance over existing methods.

## 1 INTRODUCTION

Diffusion models are a powerful class of generative models, which have been applied to a variety of domains: from static images (Ho et al., 2020; Dhariwal & Nichol, 2021) to sequential tasks such as video prediction (Ho et al., 2022b; Voleti et al., 2022; Höppe et al., 2022) and robot control (Chi et al., 2023). Models for sequential prediction are typically autoregressive, iteratively predicting the next observations given sequence history. One of the main challenges with diffusion models is the time it takes to generate a sample (Song et al., 2020a). This problem is exacerbated in sequential tasks, where generation must happen at every time step.

The main component in all diffusion models, both for static and sequential tasks, is a learned denoising process. When generating images, noise is iteratively removed by a denoising process; for sequential tasks the denoising process is conditioned on the history, and at each time step of the sequence the denoising process generates the next observation. Traditionally, a deep neural network is learned as the denoising process, while the noise process is Gaussian, making it both easy to sample and amenable to theoretical analysis (Sohl-Dickstein et al., 2015; Ho et al., 2020). Recently, however, several studies explored alternative diffusion models which use arbitrary noise processes, such as blending an image from the data distribution with an image from some source distribution (Bansal et al., 2022; Delbracio & Milanfar, 2023). In particular, Heitz et al. (2023) showed that under suitable conditions, such iterative blending and deblending (termed *iterative $\alpha$-(de)blending*) converges to the true data distribution, similarly to standard diffusion models. However, to the best of our knowledge, current state of the art results for static image generation are still based on Gaussian noise, while for sequential tasks the use of non-Gaussian noise has not been explored.

In this work we extend the iterative $\alpha$-(de)blending approach of Heitz et al. (2023), applying it to sequential tasks. Specifically, our diffusion model, which we term **A**ccelerating **S**equence **P**rediction with **De**terministic **D**enoising **D**iffusion Models (ASPeeD), learns to deblend an observation from a source distribution to the next observation (or observations) in the sequence. The key component in ASPeeD, however, and the focus of investigation in this work, is *the form of the source distribution*. We hypothesize that by designing the source distribution appropriately, we can facilitate the task of deblending. Intuitively, if the source distribution is 'more similar' to the data distribution, it is natural to expect that transforming a sample from the source to the data should be simpler. Indeed, we find that this is the case in practice. More importantly, we investigate *how* to design suitable source distributions. Our insight is that in sequential tasks where temporal changes are not too

abrupt, previous observations can be used to design a source distribution approximately similar to the true distribution of the next sample in the sequence; we propose several such designs in our work.

Empirically, we use ASPeeD in two nontrivial sequential tasks: video prediction and robot control[1]. We find that with our choice of source distributions, ASPeeD is able to sample the next observation with *significantly fewer inference steps* than conventional diffusion models and sampling acceleration methods (Song et al., 2020a), without loss of quality (and in some cases with even better quality). These results suggest that choosing the noise distribution for diffusion models – a design point that has mostly been overlooked in the literature – can have important practical benefits. In addition, to the best of our knowledge, this is the first demonstration that deterministic diffusion models based on (de)blending can significantly improve upon conventional diffusion models.

## 2 BACKGROUND

In this section, we provide background and notation for the diffusion models used in this paper.

### 2.1 DIFFUSION MODELS

Diffusion models (Ho et al., 2020; Song et al., 2020b) are generative models trained to iteratively denoise samples from a target distribution to which noise was added. A forward diffusion process gradually adds Gaussian noise to a sample over $T$ steps. A noise prediction model $\epsilon_\theta(x, t)$ is trained to identify this noise. The trained diffusion model is utilized to invert the forward process by predicting the noise at each time step $t$ given $t, x_t$. Starting from Gaussian noise, the model removes the noise gradually to generate clean samples using a sampling algorithm. Two sampling algorithms are commonly used:

**Denoising Diffusion Probabilistic Models (DDPM, Ho et al. 2020):** The refined sample at each step of the reverse diffusion process is produced by removing noise from the previous sample $x_t$ and adding a smaller amount of noise using the model $\epsilon_\theta(x, t)$. This process is time-consuming, as it requires removing noise iteratively.

**Denoising Diffusion Implicit Models (DDIM, Song et al. 2020a):** Instead of gradually adding noise, at each time step an approximation of the denoised sample, $\hat{x}_0$, is derived from $x_t$ and the model $\epsilon_\theta(x_t, t)$. Then, $x_{t-\delta}$ is approximated from $\hat{x}_0$ and $\epsilon_\theta(x_t, t)$, where $\delta \in \{t - 1, t - 2, ..., 0\}$. Intuitively, DDIM speeds up the sampling process by "skipping" some of the diffusion steps, while trading off sample quality. DDIM is used in many large scale applications such as Imagen Video and Stable Diffusion (Ho et al., 2022a; Rombach et al., 2022).

### 2.2 DETERMINISTIC DIFFUSION MODELS

Deterministic diffusion models offer an alternative formulation for denoising. A recent example of such a method is **Iterative $\alpha$-(de)Blending (IADB, Heitz et al. 2023)** which learns a deterministic mapping between two densities. The densities are linearly interpolated (blended) and a neural network model learns how to deblend them. Given source and target densities $p_0, p_1 : \mathbb{R}^d \to \mathbb{R}^+$, respectively, and samples $(x_0, x_1) \sim p_0 \times p_1$, the forward process is defined as: $x_{\alpha_t} = (1 - \alpha_t)x_0 + \alpha_t x_1$, where $\alpha_t \in [0, 1]$ is the time-dependent blending parameter. Given a blended sample $x_\alpha$, a NN model $\epsilon_\theta(x_\alpha, \alpha)$ is trained to approximate $x_1 - x_0$ by minimizing $\|\epsilon_\theta(x_\alpha, \alpha) - (x_1 - x_0)\|^2$. During inference, the model starts with $x_0$ and iteratively removes the noise until reaching $x_1$: $x_{\alpha_{t+1}} = x_{\alpha_t} + (\alpha_{t+1} - \alpha_t)\epsilon_\theta(x_\alpha, \alpha)$, where $x_{\alpha_T} = x_1$ and $x_{\alpha_0} = x_0$.

## 3 PROBLEM FORMULATION

Sequence prediction is the task of predicting the next data points in a series of sequential data (such as a video or robot trajectory), given a subset of the previous data points. Denote a state (or frame)

---

[1]Visual results are available at https://sites.google.com/view/aspeed-iclr2024.

at time $j$ as $s^j$. We attempt to predict the next series of $d$ states $\tau^{j+1,j+d} = \{s^{j+1}, \ldots, s^{j+d}\}$ given a history of the $k$ previous states of the sequence $\tau^{j-k,j} = \{s^{j-k}, \ldots, s^j\}$ [2].

Diffusion models have been used for sequence prediction by treating a trajectory of states as a sample from the target distribution (Ho et al., 2022b; Höppe et al., 2022; Chi et al., 2023; Ajay et al., 2023). We frame the problem of sequence prediction as sampling from a deterministic deblending diffusion model. We aim to recover a distribution $q : \mathbb{R}^{n \times d} \to \mathbb{R}^+$ representing target sequences of length $n$. As is common practice in sequence prediction methods, we condition this distribution on available contextual information such as the history of previous states, which we denote by $C$. Future state predictions are sampled from the target distribution as $\tau^{t,t+n} \sim q(\tau|C)$. We initialize the deblending process from a source distribution $\tilde{q} : \mathbb{R}^{n \times d} \to \mathbb{R}^+$. In contrast to standard diffusion methods, we do not limit this distribution to be Gaussian; rather, any distribution of the same dimension as the target distribution can be used. Source data samples are generated as $\tau \sim \tilde{q}(\tau|C)$ [3]. The source distribution is conditioned on the same information as the target. We denote samples from the source distribution as $\tau_0$ and samples from the target distribution as $\tau_1$.

## 4 METHOD

In this section we extend iterative $\alpha$-(de)blending to sequential prediction and show that it converges to the conditional distribution of the data (Sec. 4.1). Then, we explore the role and importance of the source distribution in deterministic diffusion. Exploiting the fact that deterministic diffusion can work with various source distributions, we ask – is there value in *designing* the source distribution to make learning easier? In Sec. 4.2 we show that indeed, the source distribution can significantly impact performance. Based on this observation, we propose several source distributions designed for sequence prediction tasks (Sec. 4.3).

### 4.1 DETERMINISTIC DIFFUSION FOR SEQUENCE PREDICTION

Given context $C$, a data sample $\tau_1 \sim q(\tau|C)$, a noisy sample $\tau_0 \sim \tilde{q}(\tau|C)$ from the source distribution and a time-dependent blending parameter $\alpha_t \in [0,1]$, the forward diffusion process of adding noise to the sample is described by a linear interpolation between the source distribution and the target distribution:

$$\tau_{\alpha_t} = (1 - \alpha_t)\tau_0 + \alpha_t \tau_1 = \tau_0 + \alpha_t(\tau_1 - \tau_0) \quad (1)$$

---

**Algorithm 1** Deterministic Deblending

**Require:** Context $C$, $\tau_0 \sim \tilde{q}(\tau|C)$, $\alpha_t$
1: For $t = 0, \ldots, T - 1$ do:
2:   $\bar{\tau}_1 = \mathbb{E}\left[\tau_1 \middle| \tau_\alpha, C\right]$
3:   $\bar{\tau}_0 = \mathbb{E}\left[\tau_0 \middle| \tau_\alpha, C\right]$
4:   $\tau_{\alpha_{t+1}} = (1 - \alpha_{t+1})\bar{\tau}_0 + \alpha_{t+1}\bar{\tau}_1$
5: Return $\tau_{\alpha_T} = \hat{\tau}_1$

---

Let $P(C)$ denote the distribution over contexts. Let $P_{C,\alpha}(\tau_\alpha)$ denote the probability distribution of the $\alpha$-blended sample (Eq. 1), conditioned on the context $C$. The minimum squared estimator $f$ of $\tau_1, \tau_0$ conditioned on the blended sample $\tau_\alpha$ is the conditional expectation (Schervish, 2012):

$$\arg\min_f \mathbb{E}_{C,\tau_1,\tau_0,\tau_\alpha}\left[(f(\tau_\alpha, C) - \tau_1)^2\right] = \mathbb{E}\left[\tau_1 \middle| \tau_\alpha, C\right],$$

$$\arg\min_f \mathbb{E}_{C,\tau_1,\tau_0,\tau_\alpha}\left[(f(\tau_\alpha, C) - \tau_0)^2\right] = \mathbb{E}\left[\tau_0 \middle| \tau_\alpha, C\right].$$

Following a similar approach to Heitz et al. (2023), we obtain the following result:

**Theorem 4.1.** *Consider the deterministic deblending algorithm (Algorithm 1). Let $P_T(\tau)$ denote the distribution of the algorithm's returned output for number of iterations $T$. For context $C$, if $\tilde{q}(\cdot|C)$ and $q(\cdot|C)$ are Riemann-integrable densities of finite variance, then*

$$\lim_{T \to \infty} P_T(\tau) = q(\tau_1|C).$$

The proof follows from the convergence theorem in Heitz et al. (2023): a stochastic deblending algorithm which replaces the expectations in lines 2 and 3 with samples from the posteriors $\bar{\tau}_1 \sim$

---

[2]In this work, indices in superscript will always be used to denote the time index of the state within the sequence; indices in subscript will always used to denote an iteration index of the diffusion process.

[3]For clarity of presentation, we drop the superscript notation for sequence time steps where unnecessary.

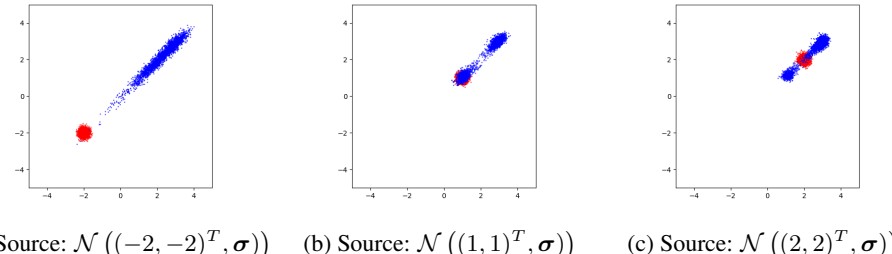

(a) Source: $\mathcal{N}\left((-2,-2)^T, \boldsymbol{\sigma}\right)$  (b) Source: $\mathcal{N}\left((1,1)^T, \boldsymbol{\sigma}\right)$  (c) Source: $\mathcal{N}\left((2,2)^T, \boldsymbol{\sigma}\right)$

Figure 1: **Source distribution affects deblending.** The target distribution in all three examples is a bi-modal Gaussian, centered at $(1,1)$ and $(3,3)$. Source distribution variance is $\boldsymbol{\sigma} = 0.1\boldsymbol{I}$. Red dots are samples from the source, blue dots are samples from the model after 10 steps of deblending.

$P\left(\tau_1 | \tau_\alpha, C\right)$ and $\bar{\tau}_0 \sim P\left(\tau_0 | \tau_\alpha, C\right)$, converging by definition to $q(\tau_1 | C)$, has the same limit as Algorithm 1 when $T \to \infty$. The only difference from Heitz et al. (2023) is the conditioning on $C$; however, since $C$ is deterministic in Algorithm 1, the proof holds as is.

Based on the above, we can use stochastic gradient descent to train a function $f$ to predict $\mathbb{E}\left[\tau_1 | \tau_\alpha, C\right]$ and $\mathbb{E}\left[\tau_0 | \tau_\alpha, C\right]$ by minimizing the mean squared error over batches sampled from the data. Alternatively, following Heitz et al. (2023), it suffices to predict $\zeta \triangleq \mathbb{E}\left[\tau_1 - \tau_0 | \tau_\alpha, C\right]$. This learning method is presented in Alg. 2, and in Alg. 3 where we show how to use the learned predictor to generate samples from $q(\tau_1 | C)$.

## 4.2 SELECTING THE SOURCE DISTRIBUTION: AN INSTRUCTIVE EXAMPLE

Since deterministic diffusion can be initialized from arbitrary sources, it may be beneficial to select these distributions carefully, aiming to simplify the deblending process and more quickly recover the target distribution. We empirically validate this hypothesis with a toy problem. We run three simple experiments[4] based on the Iterative $\alpha$-(de)blending algorithm, similar experiments based on Rectified Flow can be found in Appendix A. In the first two, we attempt to recover a target bi-modal Gaussian distribution, centered at $(1,1)$ and $(3,3)$ with variance $\boldsymbol{\sigma} = 0.1\boldsymbol{I}$, with equal probability. In both cases, we use the deterministic deblending training and inference algorithms (Alg. 2 and Alg. 3), with the model $\epsilon_\theta$ represented by a 3-layer MLP.

---

**Algorithm 2** Training Algorithm

**Require:** Context $C$, $\tau_0 \sim \tilde{q}(\tau | C)$, $\tau_1 \sim q(\tau | C)$, $t \sim Uniform(\{1, ..., T\})$
1: $\tau_{\alpha_t} = (1 - \alpha_t)\tau_0 + \alpha_t \tau_1$
2: Take gradient descent step on
3: $\quad \nabla_\theta \|\epsilon_\theta(\tau_{\alpha_t}, t) - \zeta\|^2$

---

**Algorithm 3** Inference Algorithm

**Require:** Context $C$, $\tau_0 \sim \tilde{q}(\tau_0 | C)$, $T$
1: **for** $t = 0, ..., T - 1$ **do**
2: $\quad \hat{\zeta} \leftarrow \epsilon_\theta(\tau_t, t)$
3: $\quad \tau_{\alpha_{t+1}} = \tau_{\alpha_t} + (\alpha_{t+1} - \alpha_t)\hat{\zeta}$
4: **end for**

---

In the first experiment (Fig. 1), we show that the location parameter of a Gaussian source distribution matters: deblending from distributions with the same variance and different initial means, it is clear that starting from a distribution more closely approximating the target (Fig. 1c) achieves better results. In the second experiment, we propose a simple method to obtain a useful source distribution. We deblend a Gaussian source for 3 steps, and then train an additional model to deblend samples from the first model towards the target distribution. Figures 2b and 2c show the source distribution and samples from the first model after 10 and 3 deblending steps, respectively. Fig. 2d shows samples from the second model, clearly demonstrating that intializing with the output of the first model, which matches the target more closely, is beneficial. Intuitively, after 3 deblending steps the distribution is no longer Gaussian, but is a closer approximation of the bi-modal target distribution. This is easier to deblend with 7 additional steps than with 10 deblending steps from the Gaussian source.

To demonstrate the benefits of conditioning the source distribution on relevant information, we conduct a third experiment (Fig. 3). We set a target distribution of a bi-modal Gaussian conditioned on a Bernouli random variable $C$: w.p. $0.5$ the bi-modal Gaussian is centered around $(-6, -6)$ and

---

[4]GIFs of the results are available on the supplementary website.

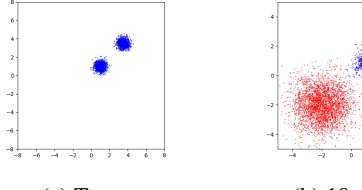
(a) Target

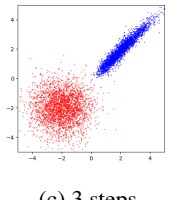
(b) 10 steps

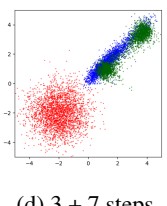
(c) 3 steps

(d) 3 + 7 steps

Figure 2: **Multi-stage deblending.** Beginning with a source distribution of $\mathcal{N}\left((-2,-2)^T, \boldsymbol{I}\right)$ (depicted by red dots), Fig. 2b and Fig. 2c show results of the deblending algorithm with 10 and 3 steps respectively towards the target bi-modal Gaussian at $(1,1)$ and $(3.5, 3.5)$ (Fig. 2a). Samples from this first model are shown in blue. Fig. 2d shows samples from the first model after 3 steps of deblending (blue), and additionally shows (in green) samples from a second model, trained to deblend the output of the first model towards the target.

$(-4, -4)$ or around $(4, 4)$ and $(6, 6)$. We train a 4-layer MLP to deblend two source distributions to this target. In Fig. 3a, an unconditional Gaussian centered around $(0, 0)$ is deblended. In Fig. 3b, the source is conditioned on the same random variable as the target, such that it is located between the modalities of the target, matching the value of $C$. It is clear that with the added condition, the deblending process is simpler and produces more accurate results.

In the next section, we describe ways of obtaining meaningful source distributions for more complex scenarios, leveraging the inherent structure of sequence prediction problems.

### 4.3 SOURCE DISTRIBUTIONS FOR SEQUENCE PREDICTION DIFFUSION MODELS

Since our deterministic deblending algorithm does not limit the source to be Gaussian, we are free to select it as we see fit. As shown in the previous section, to recover deblended samples quickly, it makes sense to initialize the deblending process with an informative source which more closely resembles the target distribution. We propose two approaches to obtain the source distribution $\tilde{q}$.

**History-based source:** In many sequence prediction domains consequent states (or frames) are similar to each other, and every state may contain information useful to predict the next. We leverage this insight by taking $\tilde{q}(\cdot|C) = \mathcal{N}(\tau^{j-k,j-1}, \gamma \boldsymbol{I})$, $k \geq 0$: the source distribution is given by a sequence of previous states, perturbed with $\gamma$-scaled Gaussian noise. For example, in video prediction, we use a single previous frame to predict the next one ($k = 0$). We name this approach **Perturbed History**. Other options for history-based source distributions exist; for instance, the perturbed output of a deterministic model predicting the next states.

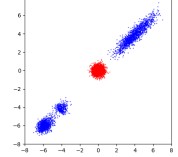
(a) Unconditional source distribution

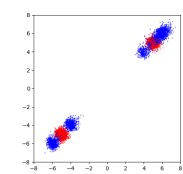
(b) Conditional source distribution

Figure 3: **Effects of conditioning.** Samples from the source (red) and deblended (blue) distributions.

**DDIM-based source:** An alternative approach is to obtain an approximation of the target distribution and continue deblending from it (similarly to the two-stage approach of the second experiment in Sec. 4.2). One way to obtain such an approximation is to use DDIM (Song et al., 2020a): $\tilde{q}(x|C) = \text{DDIM}_n(\cdot|C)$, where $\text{DDIM}_n$ denotes $n$ iterations of DDIM sampling. This approach requires a pre-trained standard diffusion model; the model is conditioned on the same context $C$, and a few steps of DDIM are taken to produce an approximation for the predicted sequence. While this approach requires training an additional model, it has the potential to greatly reduce the number of deblending steps required at inference time. We verify this assumption in our experiments.

## 5 EXPERIMENTS

We evaluate our method on two different domains: video prediction and robot control using diffusion policies. We show that our method can operate when conditioned on different contexts (robot state in the robot control tasks and images in video prediction) and using different conditioning methods

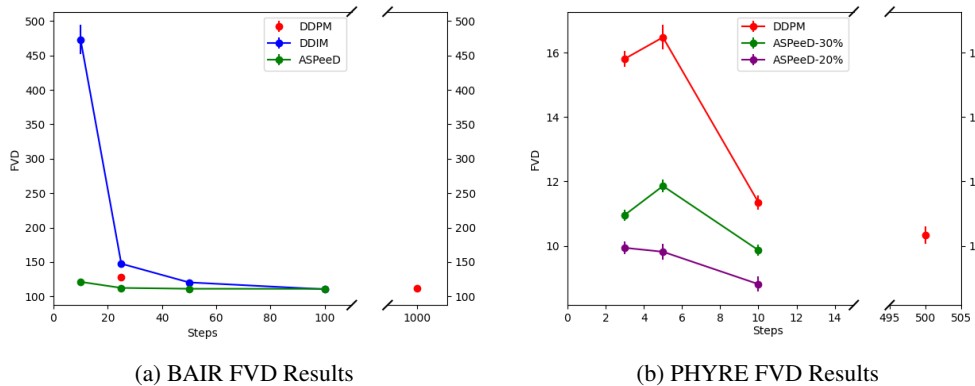

(a) BAIR FVD Results         (b) PHYRE FVD Results

Figure 4: **FVD Results.** On both BAIR (Fig. 4a) and PHYRE (Fig. 4b) datasets, ASPeeD obtains lower FVD scores than the baselines with faster inference. We omit the values of DDIM for the PHYRE dataset, as they were significantly worse than the values of the rest of the baselines.

(appending the context in video prediction and FiLM conditioning (Perez et al., 2018) in robot control). We show that our method accelerates inference while maintaining sample quality or improving it in all of the tested domains. Additional details for all experiments are available in the Appendix; video results can be viewed at `https://sites.google.com/view/aspeed-iclr2024`.

## 5.1 VIDEO PREDICTION

We test our method on a video prediction task using two different datasets. We compare our method to two baselines: a standard diffusion model using DDPM sampling for generation, and the same model with DDIM sampling. Both our model and the baseline model are conditioned on history by appending previous frames to the source sample. At each step, a single frame is predicted and appended to the history to predict the next frame autoregressively, as done in Voleti et al. (2022). While the baseline methods start denoising from Gaussian noise, the ASPeeD deblending process uses the Perturbed History distribution with $k = 0$ for BAIR and PHYRE (see Sec. 4.3). For an additional ablation study on the source distribution initialization please see Appendix D.1. To quantitatively evaluate prediction, we use the Fréchet Video Distance (FVD, Unterthiner et al. 2018), which captures semantic similarity and temporal coherence between video frames.

**BAIR Robot Pushing Dataset (Ebert et al. 2017)**: The dataset is comprised of short video clips, each with a resolution of $64 \times 64$, showcasing a robot arm interacting with various objects. We evaluated our algorithm with the same setting as Rakhimov et al. (2021), predicting the next 15 frames given a single observed frame as context. We train the baseline diffusion models with 25 and 1000 steps of DDPM, and use DDIM with different numbers of sampling steps on the 1000-step trained model to report the FVD (Fig. 4a). With ASPeeD, we take a total of 25 deblending steps from the Perturbed History source distribution.

With only 25 inference steps, ASPeeD achieves a competitive FVD to that of 100 DDIM steps taken from a model trained with 1000 DDPM steps. Surprisingly, we found that training a model with 25 DDPM steps outperforms 25-step DDIM over a 1000-step DDPM model, in contrast with well-known DDIM results (Song et al., 2020a). We found this result to be consistent with other datasets we evaluated. This finding, which is of independent interest, may hint that in sequential models the prevalent use of DDIM may be unnecessary even with Gaussian noise models. In any case, ASPeeD outperforms both DDIM and DDPM. Qualitative results of generated videos can be seen on the website and in Appendix D.2. DDIM with 25 steps generates videos where the robot moves unevenly, while our algorithm better matches the dataset and generates smoother videos.

**PHYRE** (Bakhtin et al. 2019) is a physical reasoning benchmark. We gather data of $64 \times 64$-pixel images from the BALL-tier assignments of the `ball-within-template` scenario. Models are conditioned on the previous 15 frames as context and generate one frame at a time. While the physics in PHYRE are deterministic, when conditioned on only 15 previous frames, missing information from earlier frames adds stochasticity. We demonstrate this in an ablation study (Fig. 5).

In Fig. 4b ASPeeD achieves better FVD scores with as little as 3 inference steps, compared to a standard diffusion model trained with 500, 10, 5 and 3 steps of DDPM (see Fig. 4b). Results with DDIM sampling were higher by two orders of magnitude, and are therefore omitted them from the plot[5]. In the supplementary website we present videos from various scenarios; videos generated by the DDPM algorithm distort objects and do not maintain their consistency, while ASPeeD generates videos of higher quality. Frame-by-frame qualitative results are available in Appendix D.3 and full details on the FVD computation can be found in Appendix C.1.1

## 5.2 ROBOT CONTROL USING DIFFUSION POLICIES

In this section, we test our method on two trajectory prediction datasets used by Chi et al. (2023), and compare its performance to their Diffusion Policy model. Both our model and Diffusion Policy are autoregressive: they output a trajectory of actions, a subset of these actions are taken in the environment, and the model is sampled again with an updated context of new observations. The episode is completed when the task goal is reached, or a maximum number of actions is taken. As recommended in Chi et al. (2023), we use a CNN-based architecture. Both the baseline and our model are conditioned on the last two observations with Feature-wise Linear Modulation (FiLM) conditioning (Perez et al., 2018). The source distribution for ASPeeD deblending is $DDIM_n(\cdot|C)$ with $n = 2$ or $n = 3$ (see Sec. 4.3), additional ablations on the source distribution are in Appendix E.

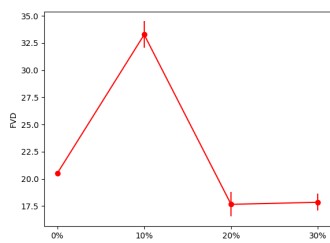

Figure 5: **PHYRE Gaussian Perturbation Ablation.** Perturbing the initial frame produces better results as it adds stochasticity. 20% perturbation results in the best FVD by a small margin over 30%.

Chi et al. (2023) present results on several robotic manipulation domains. Due to limited compute[6], we focused on two representative tasks: Push-T and Robomimic Tool-Hang.

**Tool-Hang**, the most difficult task in the Robomimic benchmark (Mandlekar et al., 2021) In this task a robot arm assembles a frame consisting of a base piece and hook piece by inserting the hook into the base, and hangs a wrench on the hook. The sparse reward is 1 when completing the task or 0 when reaching the maximum number of actions without completing it. Additional details can be found in Appendix C.2.

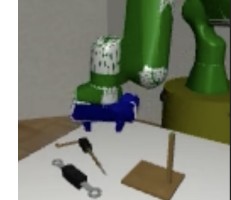

**Push-T** (Florence et al., 2021)) requires pushing a T-shaped block (grey, see Fig. 7) to a fixed target (green). End-effector (blue) and block initial positions are randomized. Reward is target area coverage percentage.

Figure 6: **Tool-Hang**

We compare ASPeeD to the Diffusion Policy baseline using MSE (measured on a held-out test set containing expert trajectories of actions) and reward (obtained by testing generated trajectories in the environment). We take a total of 10 deblending steps with ASPeeD during inference.

### 5.2.1 RESULTS

In Table 1 we compare our algorithm with DPM-Solver++ (Lu et al., 2022a), Consistency Models (Song et al., 2023) DDIM and DDPM samplers (over a 100-step DDPM model). Details of the baselines can be found in Appendix C.2. In the Tool-Hang task ASPeeD was run with 3 DDIM steps for initialization and 7 deblending steps, while in the Push-T task ASPeeD was run with 2 DDIM steps for initialization and 8 deblending steps. In total, all algorithms use 10 steps, except 100 DDPM which uses 100. Our approach is superior to all other baselines both in reward and MSE.

The MSE results show that ASPeeD learns to better imitate the expert trajectories; this advantage is also reflected in the evaluation reward (although with a smaller yet statistically significant magnitude, see Appendix B). Additionally, our algorithm adds stability: the variance over training seeds is lower both in MSE and reward compared to the baselines. We believe that the improved trajectory

---

[5]Additional comparisons to ASPeeD with history-based initialization can be found in Appendix D.1. The ablation shows that in video prediction the previous frame is a good approximation of the next frame; there is no benefit for $DDIM_n$ initialization.

[6]Each experiment takes days on an Nvidia A100 GPU; we report run times in Appendix C.2.

|  | Tool Hang | | Push-T | |
|---|---|---|---|---|
|  | Reward | MSE | Reward | MSE |
| 100 DDPM | $0.5128 \pm 0.0276$ | $0.0226 \pm 0.00143$ | $0.9156 \pm 0.0136$ | $0.0200 \pm 0.0026$ |
| Consistency-Model | $0.2613 \pm 0.0164$ | $0.0225 \pm 0.0025$ | $0.58 \pm 0.0219$ | $0.0411 \pm 0.0024$ |
| 10 DDPM | $0.5862 \pm 0.0236$ | $0.0188 \pm 0.0025$ | $0.9113 \pm 0.0267$ | $0.0199 \pm 0.0029$ |
| 10 DDIM | $0.4255 \pm 0.0288$ | $0.023 \pm 0.0021$ | $0.8725 \pm 0.0356$ | $0.0205 \pm 0.0023$ |
| DPM-Solver++2M | $0.4497 \pm 0.0431$ | $0.0229 \pm 0.0014$ | $0.9039 \pm 0.018$ | $0.0215 \pm 0.0030$ |
| DPM-Solver++3M | $0.5347 \pm 0.0574$ | $0.0232 \pm 0.0017$ | $0.8977 \pm 0.0286$ | $0.0215 \pm 0.0029$ |
| **ASPeeD** | $\mathbf{0.6150 \pm 0.0127}$ | $\mathbf{0.01410 \pm 0.0010}$ | $\mathbf{0.9273 \pm 0.0259}$ | $\mathbf{0.0174 \pm 0.0014}$ |

Table 1: **Results for robot control tasks.** Mean and STD of 10 seeds for Push-T and 3 for Tool-Hang. ASPeeD takes 10 inference steps, composed of $DDIM_2$ and $DDIM_3$ followed by 8 and 7 deblending steps respectively. It outperforms Diffusion Policies sampled by DDPM, DDIM, DPM-Solver++ and a Consistency Model with 10 steps, in addition to 100 DDPM steps.

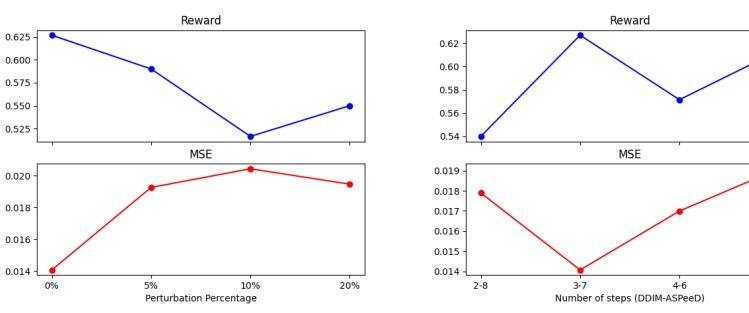

(a) Percentage of noise perturbation        (b) Number of DDIM / deterministic steps

Figure 9: **Robot control ablations.** Rewards (top) and MSE (bottom) for both our ablation experiments using a $DDIM_n$ source distribution on the Tool-Hang domain. Fig. 9a: different variance of added Gaussian noise. Fig. 9b: Different numbers of DDIM steps and deblending steps.

prediction will become more important for scaling Diffusion Policies to more difficult tasks, where prediction errors may have a higher impact on performance.

We observe that ASPeeD retains the multi-modal nature of trajectory predictions, despite being initialized with a non-Gaussian distribution. See Fig. 8 for trajectories sampled following the experiment in Chi et al. (2023), generating actions by rolling out 40 steps for the best-performing checkpoint. The stochasticity in the source distribution ($DDIM_n$, see Sec. 4.3) captures the multi-modal nature of the data. Videos are available on the website.

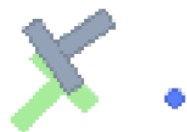

Figure 7: **Push-T**

### 5.2.2 ABLATIONS

We examine two ablations of the Tool-Hang task. First, we explore the benefit of adding Gaussian perturbation to the approximation produced by $DDIM_n$. Second, we varied the number of DDIM and deterministic steps taken (maintaining a 10-step total). Additionally, we provide an ablation study of the source distribution initialization for the Push-T task in Appendix E.

**Gaussian perturbation:** We find that perturbing the $DDIM_n$ source distribution is not beneficial to performance (see Fig. 9a). This suggests that the stochasticity of the data is captured by the DDIM sampling procedure, and that subsequent inference steps should be deterministic.

**Different ratios of DDIM to deblending:** We find that balancing the number of DDIM steps and deblending steps correctly is crucial for performance, both in MSE and rewards (see Fig. 9b). With $DDIM_2$ the initial approximation may not be good enough, and the deterministic deblending struggles to compensate for it. A good approximation as an initial guess ($DDIM_5$) without enough deterministic steps also leads to inferior outcomes.

## 6 RELATED WORK

Diffusion models have been used in a wide variety of domains as powerful generative models, most prominently in image generation (Ho et al., 2020; Dhariwal & Nichol, 2021; Song et al., 2020a; Ramesh et al., 2022). Of particular interest to our work, diffusion models have been applied to a variety of sequence prediction tasks such as video prediction (Höppe et al., 2022; Voleti et al., 2022; Yin et al., 2023; Ho et al., 2022b; Harvey et al., 2022; Qiu et al., 2019; Yang et al., 2022a;b) and for prediction of robot trajectories (Janner et al., 2022; Chi et al., 2023; Ajay et al., 2023).

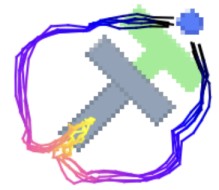

Figure 8: **Multi-modal behavior.** ASPeeD creates diverse trajectories and commits to them.

One detriment to the usability of diffusion models is inference time, caused by the inherent sequential nature of the denoising process (Song et al., 2020a). Many approaches have attempted to alleviate this issue; most notably, DDIM (Song et al., 2020a) generalizes the denoising algorithm of DDPM (Ho et al., 2020) to a non-Markovian diffusion process. Both Lu et al. (2022a;b); Zhang & Chen (2022) take advantage of the semi-linear property of the diffusion ODE to use more accurate ODE solvers; Song et al. (2023) recognize the significance of a consistent prediction along a trajectory, while Karras et al. (2022) explore design choices in diffusion algorithms. All these methods trade sample quality for sampling speed. Salimans & Ho (2021) develop a method to distill trained diffusion models and reduce the number of steps required to generate new samples.

Other, more recent work aims to perform denoising diffusion starting with non-Gaussian noise. Bansal et al. (2022) propose Cold Diffusion, which aims to learn the reverse process of non-Gaussian image transformations such as blurring or downsampling. Delbracio & Milanfar (2023) take a similar iterative approach to image restoration. Lyu et al. (2022) propose ES-DDPM, which uses samples from a pre-trained model such as a GAN or VAE as a starting point for the denoising process. Closely related to our work are Iterative $\alpha$-(de)Blending (IADB, Heitz et al. 2023) and Rectified Flow (Liu et al., 2022). Both provide recipes for blending between two arbitrary distributions. While able to improve on inference time, the above approaches center almost exclusively on image generation. In this work we focus on sequence prediction, utilizing the inherent properties of sequences to obtain better initial approximations for the denoising process. An extended related work section can be found in Appendix F.

## 7 CONCLUSION AND OUTLOOK

We present ASPeeD, a deterministic diffusion method for sequence prediction, focusing on the role of the source distribution, its importance and methods for its design. ASPeeD requires only minor changes to standard diffusion models. We demonstrate the efficacy of ASPeeD on four tasks in two distinct domains, accelerating inference over baselines while retaining or improving sample quality.

**Limitations:** While ASPeeD improves on standard diffusion models in terms of inference speed, this is traded off with training time: when training with the $DDIM_n$ source distribution, each forward pass requires sampling with DDIM as the initial approximation, and thus takes longer. In addition, when using DDIM access to a pre-trained standard diffusion model is required. In our work, we use the same models as the DDPM baselines; however, using ASPeeD may require training an additional model. Finally, ASPeeD adds hyperparameters to standard diffusion: ratios of DDIM to deblending steps when using $DDIM_n$, and the scale of perturbation noise when using history-based source distributions.

**Future Work:** We plan to extend the evaluation of ASPeeD to additional domains, including trajectory planning for real robots. These extensions may require more principled approaches to the selection of source distributions, which is an interesting line of exploration for future work. In addition, in our work we note discrepancies between datasets in the optimal number of DDPM steps required to denoise samples; the relationship between dataset properties and sampling with diffusion may merit further investigation.

## REPRODUCIBILITY

As ASPeeD is a simple modification to existing diffusion methods, much of the code used to produce the results in this paper can be found in open-source repositories. We base the implementation of our method on open-source code from `https://github.com/lucidrains/denoising-diffusion-pytorch` for the video prediction tasks, and `https://github.com/real-stanford/diffusion_policy` for the robot control tasks, with minor modifications. All hyperparameters are reported in Appendix C. In addition, we intend to release our source code upon acceptance of the paper.

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

APPENDIX

## A    INSTRUCTIVE EXAMPLES WITH RECTIFIED FLOW

We ran the three instructive examples described in the paper in Section 4.2 based on the Rectified Flow algorithm (Liu et al., 2022) instead of Iterative $\alpha$-(de)blending (Heitz et al., 2023): Experiment 1 (Fig.10), Experiment 2 (Fig.11) and Experiment 3 (Fig.12). Additional videos of the results are available on the anonymous project website. The results show that correct selection of the initial distribution is advantageous for improved performance, regardless of whether the underlying algorithm is Rectified Flow or Iterative $\alpha$-(de)blending. In the videos, it can be seen that the results using Rectified Flow move in straighter lines compared to Iterative $\alpha$-(de)blending results, though the resulting distributions seem similar. These results are based on three reflow operations ($N = 3$) (see Liu et al. (2022) for details).

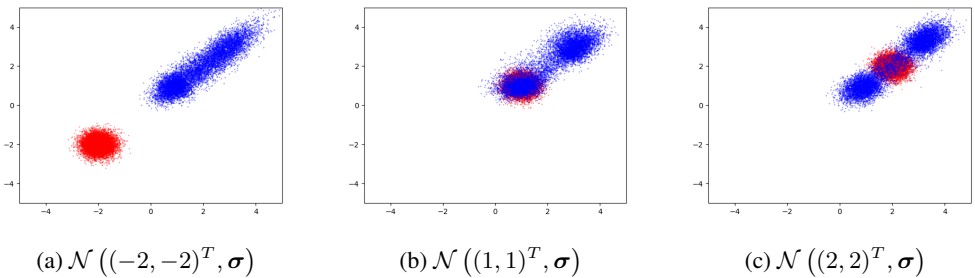

(a) $\mathcal{N}\left((-2,-2)^T, \boldsymbol{\sigma}\right)$     (b) $\mathcal{N}\left((1,1)^T, \boldsymbol{\sigma}\right)$     (c) $\mathcal{N}\left((2,2)^T, \boldsymbol{\sigma}\right)$

Figure 10: **Effects of Rectified Flow source distribution.** Source distribution for each example is described in its caption. The target distribution in all three examples is a bi-modal Gaussian, centered at $(1,1)$ and $(3,3)$. Source distribution variance is $\boldsymbol{\sigma} = 0.1\boldsymbol{I}$. Red dots are samples from the source, blue dots are samples from the model after 10 steps of the Rectified Flow process.

## B    STATISTICAL SIGNIFICANCE OF ROBOT CONTROL RESULTS

We performed an unpaired T-test over the rewards in the Diffusion Policy domain (Fig. 1) for both Tool-Hang and Push-T to validate statistical significance. In both tasks the test compared ASPeeD and the baseline with the closest results: the 10-step DDPM model. For each algorithm we chose the best checkpoint of each training seed to create a total of 4000 samples for Push-T and a total of 6000 samples for Tool-Hang. In both tasks we found that our algorithm provides a statistically significant

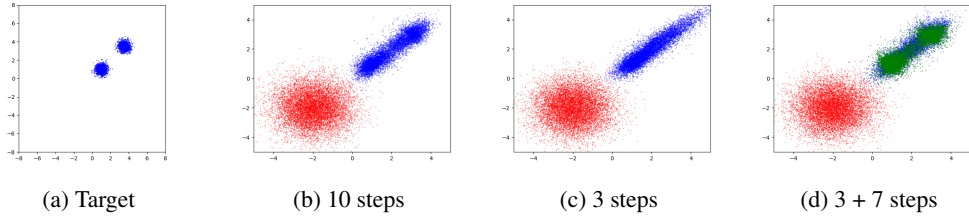

(a) Target             (b) 10 steps             (c) 3 steps             (d) 3 + 7 steps

Figure 11: **Rectified Flow multi-stage blending.** Beginning with a source distribution of $\mathcal{N}\left((-2,-2)^T, \boldsymbol{I}\right)$ (depicted by red dots), Fig. 11b and Fig. 11c show results of the Rectified Flow algorithm with 10 and 3 steps respectively of sampling towards the target bi-modal Gaussian at $(1,1)$ and $(3.5,3.5)$ (shown in blue in Fig. 11a). Samples from this first model are shown in blue. Fig. 11d shows samples from the first model after 3 steps of Rectified Flow (blue), and additionally shows (in green) samples from a second model, trained to start the Rectified Flow process from the output of the first model towards the target.

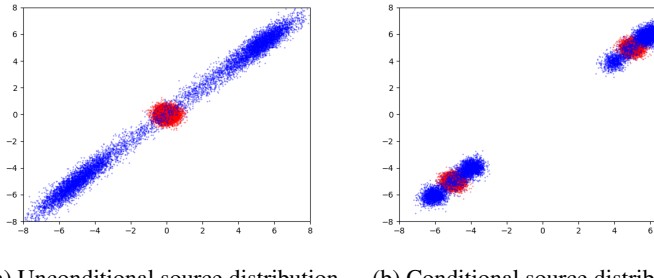

(a) Unconditional source distribution   (b) Conditional source distribution

Figure 12: **Effects of conditioning with Rectified Flow.** Samples from the source (red) and target (blue) distributions. The source is conditioned on the same random variable as the target, such that it is located between the modalities of the target. It is clear that with the added condition, the Rectified Flow sampling is simpler and produces more accurate results.

improvement in performance, with values of ($p$-value : 0.00126, $t$-value : 3.226) for Tool-Hang and ($p$-value : 0.00156, $t$-value : 3.165) for Push-T.

## C  IMPLEMENTATION DETAILS

### C.1  VIDEO PREDICTION

We implemented a 2D-convolutional U-net based on the code of `https://github.com/lucidrains/denoising-diffusion-pytorch` for the ASPeeD and the baselines, all sharing the same architecture. We used a cosine scheduler for $\alpha_t$ as common with diffusion models.

|  | BAIR | PHYRE |
|---|---|---|
| Number of Conditioned Frames | 1 | 15 |
| Number of Generated Frames | 1 | 1 |
| Batch Size | 64 | 64 |
| Learning Rate | $2e-5$ | $2e-5$ |
| U-net Dimension Multiplications | [1,2,4,8] | [1,2,4,8] |
| Number of Gradient Steps | 2M | 1.1M |
| Training Time (on a single A100 GPU) | 140 Hours | 48 Hours |

Table 2: **Video Prediction Hyperparameters.**

#### C.1.1  PHYRE FVD

We generated the PHYRE dataset with $64 \times 64$ image resolution for all available tasks from the BALL-tier assignments of the `ball-within-template` scenario in (Fig. 4b). For FVD evaluation on PHYRE we predicted 50 frames (conditioned on 15 frames) and generated 100 videos for each video in the test dataset. The test dataset has a total of 500 videos.

### C.2  ROBOT CONTROL USING DIFFUSION POLICIES

We trained all the models, sharing the same architecture, using the code provided by Chi et al. (2023) with the following parameter modifications:

As with the original model, for each task the model output is 16 actions times the state dimension (depends on the task). Out of the predicted 16 actions only the first 8 are taken in the environment, and the model is called again with updated context of observations from the environment until the task is completed or the maximum number of actions is reached.

For evaluation (Fig. 1), we chose the best checkpoint for each seed and used it to generate trajectories.

|  | Push-T | Tool-Hang |
|---|---|---|
| $n$ (for the $DDIM_n$ source distribution) | 2 | 3 |
| Number of Deblending Steps | 8 | 7 |
| Batch Size | 256 (same as original) | 1028 |
| Epochs | 5000 (same as original) | 20000 |
| Number of Training Seeds | 10 | 3 |
| Compute (on a single A100) | 10 Hours | 58 Hours |

Table 3: **Robot Control Hyperparameters.** The compute is reported for one training seed.

MSE: we compute the test MSE (for a held-out set of trajectories the model did not see during training time) as an average of each training seed over 100 environments (initial conditions), generating 10 sampled trajectories in each environments.

Reward: For Tool-Hang, the reported reward is an average over the 10 training seeds, 8 testing seeds where in each testing seed 50 environments are created and in each environment 5 trajectories are sampled. For Push-T, the reported reward is an average over the 3 training seeds, 8 testing seeds where in each testing seed 50 environments are created and in each environment 10 trajectories are taken.

|  | Push-T | Tool-Hang |
|---|---|---|
| Number of Objects | 1 | 2 |
| Action Dimension | 2 | 7 |
| Max Actions | 300 | 700 |
| Number of Demonstrations | 200 | 200 |
| High Precision Requirement | Yes | Yes |

Table 4: **Simulation Benchmark.** Both tasks have only proficient human demonstrations and not multi-human demonstrations. See Chi et al. (2023) for additional details on the dataset.

**Baseline details for the results in Table 1:** We compare our model to DPM-Solver++ (Lu et al., 2022a) of order 2 and 3 with the Multistep solver. In all runs we used the DPM-Solver++ parameters suggested in the official Github repository. We also compare to Consistency Models (Song et al., 2023) with 10 sampling steps (we note that 10 sampling steps of the Consistency Model baseline performs slightly better than 1 sampling step). We implemented Consistency Models on top of the Diffusion Policy code according to Algorithm 3 (CT) described in their paper, using L2 loss which we used for all other runs as well.

# D  ADDITIONAL VIDEO PREDICTION DETAILS

In this section, we provide ablation study of the source distribution as well as qualitative results for the video prediction experiments (Sec. 5.1).

## D.1  SOURCE DISTRIBUTION ABLATION

Fig. 13 is an extension of the FVD plot for the PHYRE dataset in Fig. 4b with the addition of experiments where the source distribution is initialized from $DDIM_n$ instead of the previous frame. We experimented with $DDIM_2$, $DDIM_3$ and $DDIM_4$ initializations, in each case following up with deblending steps to complete a total of 5 or 10 steps.

The experiments conducted are as follows:

- 2 DDIM steps (Blue) followed by 8 or 3 deblending steps, bringing the total to 10 or 5 steps respectively.

- 3 DDIM steps (Orange) followed by 7 or 2 deblending steps, bringing the total to 10 or 5 steps respectively.

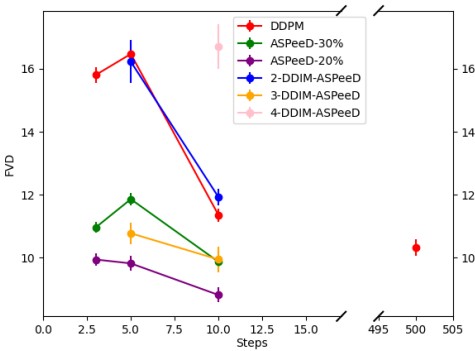

Figure 13: PHYRE FVD (lower is better) with additional results of $DDIM_n$ initialization. In blue the initialization is $DDIM_2$ followed by 3 or 8 deblending steps, in orange the initialization is $DDIM_3$ followed by 2 or 7 deblending steps, in pink the initialization is $DDIM_4$ followed by 6 deblending steps. History-based initialization outperforms $DDIM_n$ initialization, with the correct amount of perturbation noise applied.

- 4 DDIM steps (Pink) followed by 6, bringing the total to 10 steps

On the plot in Fig. 13, each result is represented by its combined total number of steps (5 or 10).

$DDIM_3$ produced better results than $DDIM_2$ for both 10 and 5 total steps. We believe this is due to the better initial prediction; although $DDIM_4$ had the worst results, we believe this is due to not enough deblending steps. However, the $DDIM_n$ initialization did not improve results on the PHYRE dataset compared to the previously presented history-based initialization. As we hypothesize in the paper, this occurs due to the nature of the task: in video prediction frames are predicted one frame at a time, and the previous frame is a good approximation of the next frame.

### D.2 BAIR FRAME-BY-FRAME RESULTS

Generated BAIR 15-frame videos conditioned on the same frame outlined in red (a total of 16 frames) can be found in Fig. 14 and Fig. 15. The complete videos can be viewed on the supplementary website. The 25-step DDIM is sampled from the 1000-step DDPM model.

### D.3 PHYRE FRAME-BY-FRAME RESULTS

Fig. 16 and Fig. 17 show generated PHYRE 50-frame videos conditioned on the same 15 frames, for a total of 65 frames. The condition (context) frames are outlined in red. The complete videos can be viewed on the supplementary website. The DDPM and DDIM based models cause object distortion and disappearance while ASPeeD is more coherent.

## E ROBOT CONTROL USING DIFFUSION POLICIES: ABLATION

In this section we provide an ablation study of the source distribution initialization for the Push-T task in Fig. 18, where we duplicated the current observation and added Gaussian perturbation (for Perturbed-History initialization). We compared Perturbed History initialization with 10 deblending steps and different levels of perturbation: $20\%, 30\%$ and $40\%$ with the $DDIM_2$ based initialization, where 2 steps of DDIM are taken followed by 8 deblending steps. The results show that in this domain $DDIM_n$ initialization is more beneficial than history based initialization as the rewards are higher and the MSE is lower. We suspect that this is due to the nature of the prediction task: the output of the network is 16 future states, so initializing from the current state is not a good approximation of the required sample. The ablation is averaged based of 3 random training seeds and 8 different testing seeds.

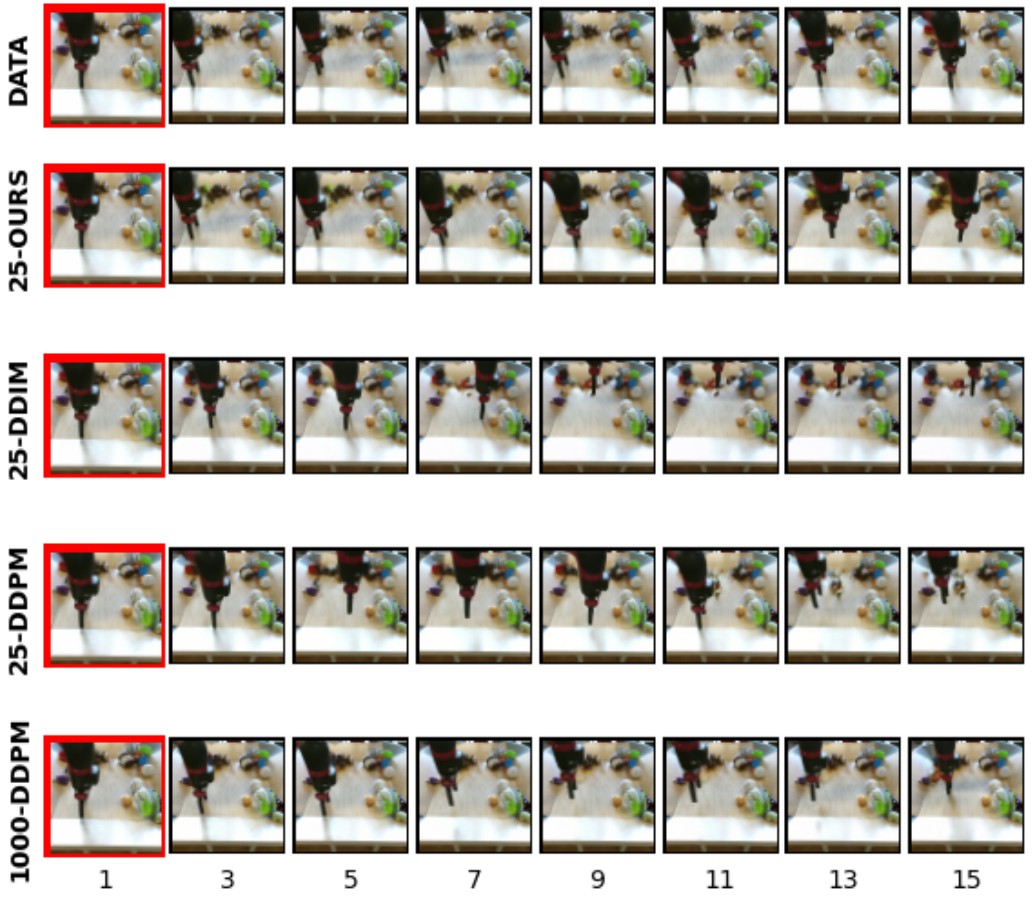

Figure 14: **BAIR Frames 1.** The frames are numbered at the bottom. Ground truth video from the dataset can be seen in the top row, our generated video in the second row and the baselines in the rest of the rows. The DDIM generated video moves at uneven speeds.

# F    EXTENDED RELATED WORK

Diffusion models have exploded in popularity in recent years, and have been used in a wide variety of domains as powerful generative models, most prominently in image generation (Ho et al., 2020; Dhariwal & Nichol, 2021; Song et al., 2020a; Ramesh et al., 2022). Of particular interest to our work, diffusion models have been applied to a variety of sequence prediction tasks. In particular, they have been used to varying degrees of success for video prediction (Höppe et al., 2022; Voleti et al., 2022; Yin et al., 2023; Ho et al., 2022b; Harvey et al., 2022; Qiu et al., 2019; Yang et al., 2022a;b), as well as for decision making and prediction of robot trajectories (Janner et al., 2022; Chi et al., 2023; Ajay et al., 2023).

One detriment to the usability of diffusion models is inference time, caused by the inherent sequential nature of the denoising process (Song et al., 2020a). Many approaches have attempted to alleviate this issue; most notably, DDIM (Song et al., 2020a) generalizes the denoising algorithm of DDPM (Ho et al., 2020) to a non-Markovian diffusion process. Both Lu et al. (2022a;b); Zhang & Chen (2022) take advantage of the semi-linear property of the diffusion ODE to use more accurate ODE solvers; Song et al. (2023) recognize the significance of a consistent prediction along a trajectory, while Karras et al. (2022) explore design choices in diffusion algorithms. All these methods

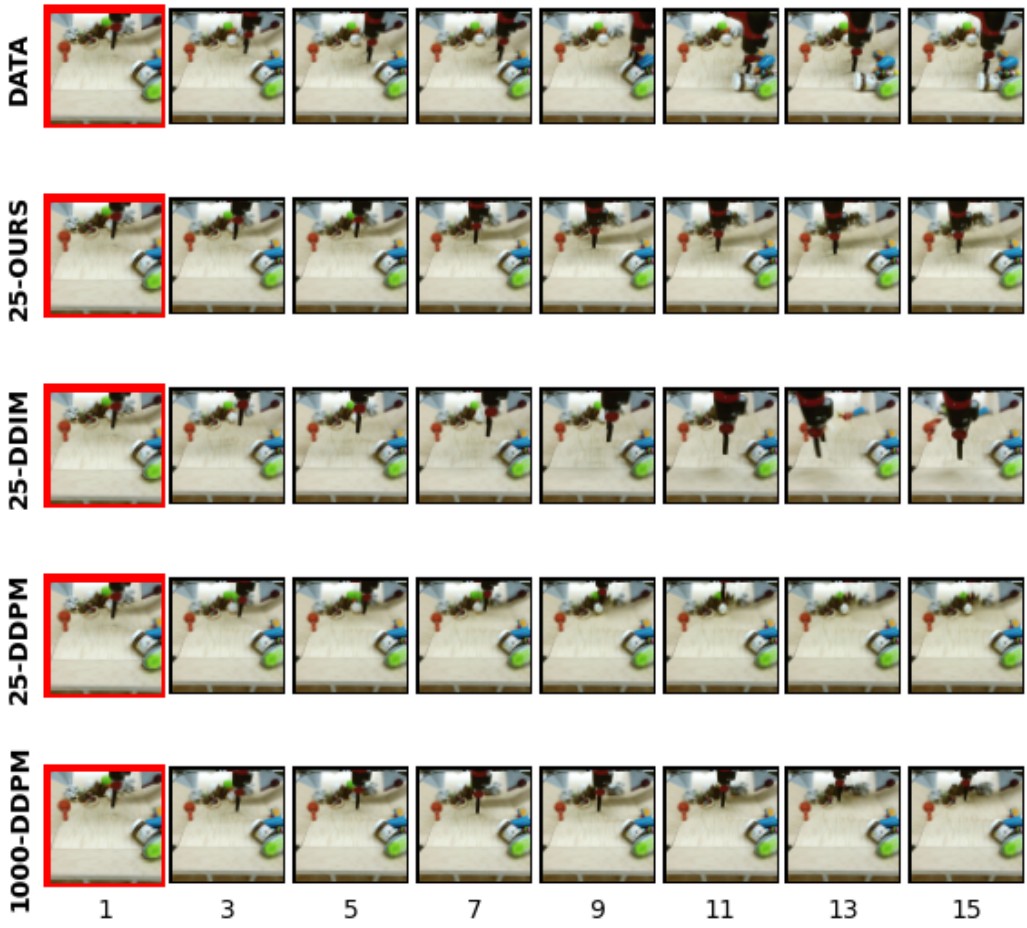

Figure 15: **BAIR Frames 2.** The frames are numbered at the bottom. Ground truth video from the dataset can be seen in the top row, our generated video in the second row and the baselines in the rest of the rows. The DDIM generated video moves at uneven speeds.

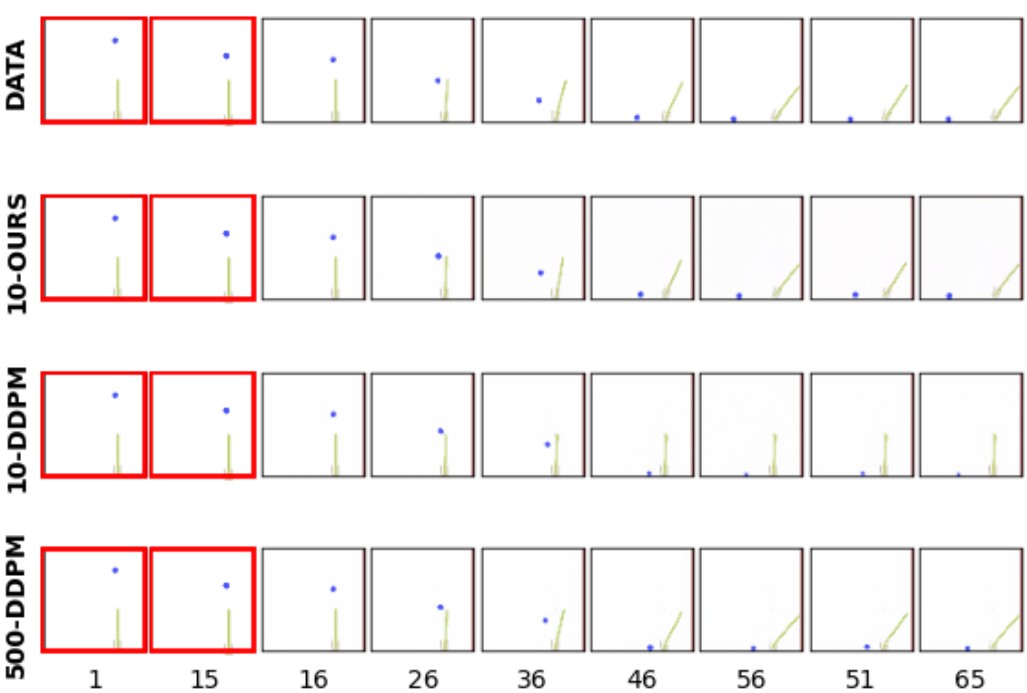

Figure 16: **PHYRE Frames 1.** The frames are numbered at the bottom. The ground truth video from the dataset can be seen in the top row, our generated video in the second row and the baselines in the rest of the rows. The frames generated by the 10 step DDPM baseline do not match the behavior in the data: the green stick does not move correctly, and in the 500 step DDPM baseline frames the green stick bends unnaturally; while our algorithm generates physically valid frames.

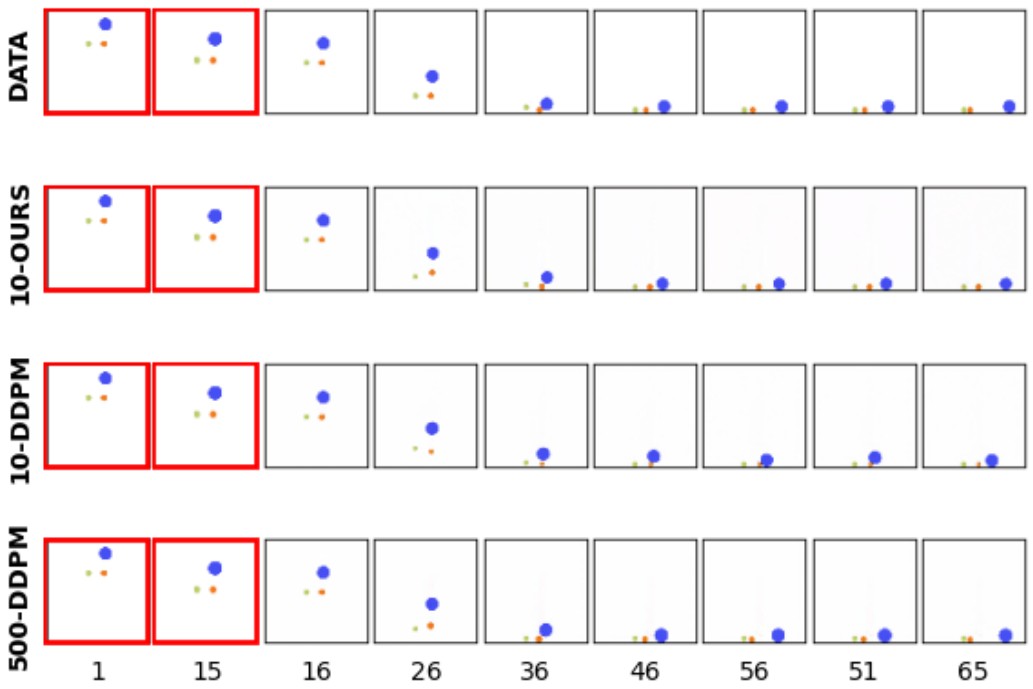

Figure 17: **PHYRE Frames 2.** The frames are numbered at the bottom. The ground truth video from the dataset can be seen in the top row, our generated video in the second row and the baselines in the rest of the rows. In the frames generated by the 10 step DDPM baseline the blue ball does not move correctly.

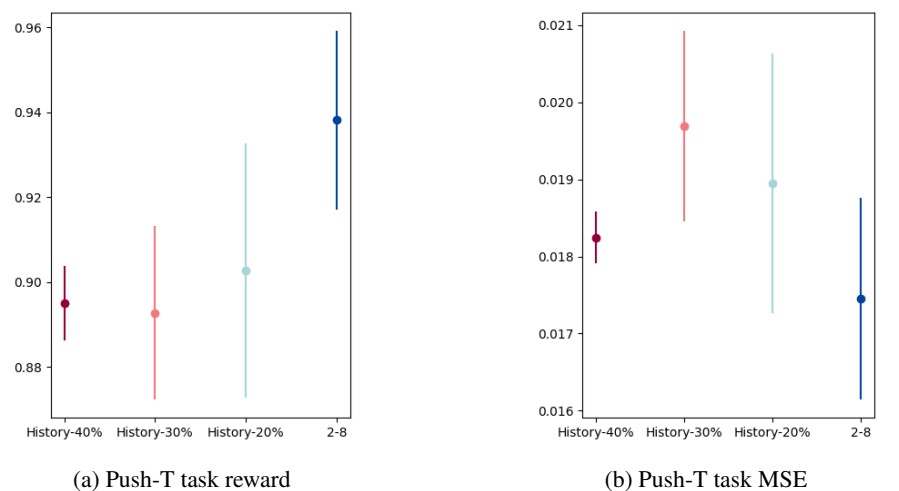

(a) Push-T task reward    (b) Push-T task MSE

Figure 18: **Robot control history-based initialization ablations. Push-T task reward and MSE comparison with different initialization techniques: history-based initialization with $20\%$, $30\%$ and $40\%$ perturbation and 10 deblending steps, and $DDIM_2$ initialization with 2 DDIM steps and 8 deblending steps (10 total). In this task $DDIM_n$ initialization is superior to history-based initialization.**

trade sample quality for sampling speed. Salimans & Ho (2021) develop a method to distill trained diffusion models and reduce the number of steps required to generate new samples.

Other work aims to perform denoising diffusion starting with non-Gaussian noise, in some cases even taking deterministic steps to denoise non-stochastic transformations. Bansal et al. (2022) propose Cold Diffusion, which aims to learn the reverse process of non-Gaussian image transformations such as blurring or downsampling. Delbracio & Milanfar (2023) take a similar iterative approach to image restoration. Lyu et al. (2022) propose ES-DDPM, which uses samples from a pre-trained model such as a GAN or VAE as a starting point for the denoising process. Closely related to our work are Iterative $\alpha$-(de)Blending (IADB, Heitz et al. 2023) and Rectified Flow (Liu et al., 2022). Both provide a recipe for blending between two arbitrary distributions. While able to improve on inference time by initializing the diffusion process from distributions other than Gaussian noise, the above approaches center almost exclusively on image generation. In this work we focus on sequence prediction, utilizing the inherent properties and available information in sequences to obtain better initial approximations for the denoising process. Lee et al. (2021) considered an audio domain, and proposed to initialize a non-deterministic diffusion process from a learned Gaussian source distribution based on the data. Though they present a novel source distribution, it is still Gaussian and the algorithm is based on the standard diffusion process. In our work we consider video prediction and robotic control, and explore *non-Gaussian* source distributions, using deterministic diffusion. In parallel, previous studies such as Denton & Fergus (2018); Walker et al. (2021) have explored prior selection for video prediction; however, Denton & Fergus (2018); Walker et al. (2021) use variational autoencoders (VAEs), while we focus on the more performant diffusion models.

