# OpenReview forum: "Deterministic Diffusion for Sequential Tasks"
_ICLR.cc/2024/Conference — Submitted to ICLR 2024_

### Official Review · Reviewer_X2Za · 2023-10-14

**Soundness:** 3 good
**Presentation:** 2 fair
**Contribution:** 2 fair
**Rating:** 5
**Confidence:** 4

**Summary:**

This paper proposes a deterministic diffusion method for video prediction and robotic control. The main claim is that training from a better distribution can lead to faster inference. The authors conduct extensive experiments on the BAIR Robot pushing dataset and PHYRE, which show that their approach can boost generation and policy learning under many conditions.

---

Post rebuttal comments: I think the authors provide a lot of useful responses, and some parts of my initial review are incorrect (have been fixed). However, after the clarification, this paper looks even more like an engineering improvement over the prior work Heiz et al. Therefore, I keep my scores unchanged.

**Strengths:**

1. The proposed method is verified on distinctive tasks and the demo videos look cool.
2. The authors provide a deterministic diffusion model for sequential tasks, which are very general. The idea of utilizing a better initialization makes sense.
3. The paper introduces various toy examples to illustrate the idea. This is a strong benefit.

**Weaknesses:**

1. This paper is not particularly novel.

(1) The major change of this paper is using a better initialization. Although such an initialization may benefit the diffusion procedure, this may not be a significant change. The main model and workflow are not changed, so the proposed initialization methods look like an engineering trick.

(2) The method of selecting the source distribution seems too trivial. When using perturbed history, the agent might get stuck in the previous information, which may harm the performance. The idea of using DDIM inversion is not novel as well.

(3) This paper does not compare to many works in speeding up diffusion models. The only compared baselines are standard diffusion models without speeding up tricks. It's better to compare to IADB or other works.

2. This paper is extensively based on Heitz et al. (2023), which is not a well-known paper yet. The proposed initialization scheme should be tested in other well-known diffusion schemes and that paper should be presented in a separate section.

**Questions:**

I don't have other questions at this moment.

---

> ### Author Response · Authors · 2023-11-19
>
> We thank the reviewer for the valuable input. We address each point separately:
>
> Novelty: See our general comment to all reviewers above. We do believe that our method of using deterministic diffusion to be able to initialize from a source distribution that is not Gaussian, and using this in the context of sequential prediction, is novel. We are not familiar with previous methods which initialize the source distribution using DDIM.
>
> Concern 1.2: The agent getting stuck in the previous information is possible with warm starting where previous information is perturbed only at inference time (see eg. this paper). We, on the other hand, **train** the model to use previous data points to reach the next data point. Thus, the agent will not be stuck, as the model is trained for this scenario. We validate this empirically, showing in Fig. 4 that our algorithm has significantly lower FVD (which is better) than DDIM. In addition, we extended the Fig. 4 experiments in the supplemental rebuttal PDF (Section B.1), adding $DDIM_n$ initialization on the PHYRE dataset. We show that this initialization also provides superior performance than the other baselines (while still being inferior to the history-based initialization).
> DDIM Inversion: With DDIM-based initialization, we use the DDIM result as the initial point for the deblending process which then attempts to reach the original data point (without reversing the order of DDIM sampling).
>
> Baselines: This is a valuable insight, which is expressed by the other reviewers as well. In the experiments described in the general comment with results in the supplementary rebuttal PDF (Sections A and C.2) we address this concern by comparing our approach to Rectified Flow, DPM-Solver++ and Consistency Models.
>
> Concern 2:  We do not understand why the reviewer claims that our approach deteriorates performance. We found performance improvements In **ALL** of our experiments. In Fig. 4 and Fig. 8 we demonstrate our method's superior performance for video prediction and robotic control, respectively. In Fig. 4 our FVD is significantly lower (which is better) and in Fig. 8 our reward is higher and our MSE is notably lower (both better). In addition, we extended our baselines in Fig. 8 and added experiments to Fig. 4 in the rebuttal PDF (Section B.1 and C.2), which show the superiority of our method over additional baselines. See our previous comments regarding baselines and concern 1.2.

---

> > ### Comment · Reviewer_X2Za · 2023-11-20
> > **Thanks for your response.**
> >
> > My comment 2 refers to Figure 10, and it's my bad not to mention this figure explicitly. The manuscript says, "We find that adding Gaussian perturbations to the DDIMn source distribution is not beneficial to performance (see Fig. 10a)." So I say, "The proposed approach often deteriorates the performance. If it is true that a better initialization may lead to better performance, the authors should find that the accuracies are improving as well. Probably, the deteriorated performance means that the carefully chosen initialization scheme is incorrect." in my review. Can you clarify this?
> >
> > Also, in Fig.4, what's the score of your model in 1000 steps?

---

> > > ### Author Response · Authors · 2023-11-20
> > >
> > > We thank the reviewer for the quick reply.
> > >
> > > Figure 10: The reviewer is correct in writing that the initialization scheme should be correct, Figure 10 supports this statement. In the $DDIM_n$ initialization the initialization itself contains stochasticity as it is created from DDIM sampling. We performed an ablation study to explore the effect of adding Gaussian perturbation to the $DDIM_n$ initialization (i.e., adding noise to the result of the DDIM sampling). Adding Gaussian noise to DDIM can be seen as an interpolation between the structured DDIM distribution and the unstructured Gaussian noise baseline. Figure 10(a) shows the results of this ablation study. The plot in the original paper and the additional ablation study in the rebuttal PDF (Figure 5) show that there is no benefit in adding any Gaussian noise to the $DDIM_n$ initialization; the stochasticity is captured completely by the initialization itself. Thus, as we expect, with more noise the performance deteriorates.
> > >
> > > Figure 4: The aim of the experiments in Figure 4 is to show that our method reaches a good score in fewer steps than other methods; hence, we did not run any experiments on the “full” amount of timesteps with our approach. On the BAIR dataset, In 25 steps we obtain approximately the same score as 1000 steps of DDPM (just one point above) and in 50 steps results are identical. On the PHYRE dataset we outperform 500 steps of DDPM in only 10 steps of our method.
> > > We can add the result of the 1000 steps of our method on the BAIR dataset experiment to the final version of our paper; due to the time constraints of the rebuttal period it will not be available in the next few days. However, we are running the experiment the reviewer suggested on the PHYRE dataset (500 steps of our method), as it may be ready before the rebuttal ends. If it is, we will update the response accordingly.
> > >
> > > Please let us know if you have further questions.

---

> > > > ### Comment · Reviewer_X2Za · 2023-11-21
> > > > **Some points are still pretty unclear. I do not understand over 50% of this paper.**
> > > >
> > > > Thanks for your response again. I read this paper again. Honestly speaking, I still don't understand this paper and I think I can only understand <50% contents of this paper, and the writing does not flow well. It seems that this paper is trying to present a simple idea: "change the Gaussian noise to a better initialization," but I have trouble understanding it in many places.
> > > >
> > > > 1. Question on Figure 10. Why add noises to your trained model? Your strongest claim seems to be in the introduction, "SOTA results are still based on Gaussian noise," but adding some noise to your trained model cannot prove that current Gaussian noise-based models are not good. I don't quite understand the meaning of Figure 10. I understand $DDIM_n$ + noise is bad, but it does not mean that noise is bad.
> > > >
> > > > 2. Question in Figure 4. Why don't you run your model for 1000 steps? Is it because your model runs slowly? Note that the gap between DDIM and ASPeed is becoming smaller, so it's unsure which method performs better at 1000 steps.
> > > >
> > > > 3. Question on the main paper:
> > > >
> > > > (1) Why not use some real figures and videos to show the idea? The visualizations are not very accessible. It seems that this paper is trying to say, "using DDIM is better than Gaussian." I think this idea is proven in other works using better visualizations. I cannot directly see the real difference if the visualizations aren't very clear.
> > > >
> > > > (2) How's the target distribution known in practice?
> > > >
> > > > (3) What place should I run the Algorithm 1? In training or testing? Do I need a pre-trained diffusion model? What's the noisy sample used here? How to get the noisy sample? Why use de-blending?  How to evaluate the expectation in a batch or a dataset? What's the function of Algorithm 1 (it's never introduced)? I think it's better not to use any algorithm block in the next version of the manuscript. It's very painful to read and understand the current draft.

---

> > > > > ### Author Response · Authors · 2023-11-21
> > > > >
> > > > > We thank the reviewer for their questions.
> > > > >
> > > > > Our presentation follows Heitz et al. (2023), and we kindly refer the reviewer to that published work for a more detailed explanation of deblending, and for the foundations of our approach.
> > > > >
> > > > > Nevertheless, here’s a simple breakdown of our paper:
> > > > > 1. Deterministic diffusion (deblending) learns to map an arbitrary source distribution to a data distribution. This is done by learning a function that, when applied iteratively to a sample from the source, brings it to be a sample from the data.
> > > > > 2. Algorithm 1 describes such a function (lines 2-4). The two equations before Thm 4.1, and the last paragraph in Sec 4.1 explain how to learn this function (actually, a slight variation of it that empirically performs better, based on Heitz et al., 2023) from data sampled from both the source distribution and data distribution. This part is rather standard.
> > > > > Thus, Algorithm 1 is a theoretical construct that establishes the soundness of our method (Thm 4.1), and Algorithms 2+3 provide an approximation that can be efficiently learned from data. We remark that this approach closely follows Heitz et al. (2023), to which we kindly refer the reviewer if it is still not clear enough.
> > > > > 3. We focus on *how* to select the source distribution. We hypothesize that a source distribution that is more similar to the data distribution would be easier to learn. This is demonstrated in Section 4.2 on a toy problem, for illustration purposes.
> > > > > 4. Focusing on two sequential prediction domains, our insight is that the past frames could be used to construct a relatively good source distribution. We provide two such examples:
> > > > > (a) Simply take the last frame and add Gaussian noise.
> > > > > (b) Train *another* diffusion model (DDPM) to predict the next frame, and take a few DDIM steps using it. This yields a coarse approximation to the next frame distribution, which does not require additional Gaussian noise.
> > > > > 5. Our main contribution is showing that our approach, i.e., using one of the coarse distributions above + a few steps of deterministic diffusion, yields very good results on two domains – video prediction and robotic control. It actually surpasses several state-of-the-art baselines that were designed for diffusion, with fewer inference steps. This is an empirical finding that we believe to be important.
> > > > >
> > > > > We hope that this clarifies our work. We next address specific questions:
> > > > >
> > > > > Figure 10: **Figure 8 already establishes that our $DDIM_n$ source distribution is better than a Gaussian source distribution**. Figure 10 further explores this using an ablation study. In this ablation, we added Gaussian noise to $DDIM_n$, which can be seen as an interpolation between the structured $DDIM_n$ source and the unstructured Gaussian source baseline. Thus, as we expect, with more noise (i.e., a source that is closer to the baseline), the performance deteriorates.
> > > > >
> > > > > Figure 4: Our model requires less sampling iterations, hence it runs faster, not slower. We didn’t run the model for 1000 steps as we wanted to show that our method achieves the same quality as 1000 DDPM sampling steps in significantly less steps – 50 steps in this case. In order to run our model for 1000 sampling steps we have to re-train it. Training any video prediction model with 1000 inference steps (either DDPM or our approach) is a time-consuming effort.
> > > > >
> > > > > Visualizations: videos are available on the anonymous [website](https://sites.google.com/view/aspeed-iclr2024), as we referenced on page 6 in the paper. There are additional frame by frame pictures in the Appendix Section A.3. On the BAIR dataset it is hard to discern any visible differences,  though the FVD (which is a common metric for video prediction quality) is significantly lower (which is better);however, on the PHYRE dataset there is a clear advantage to our method as with the DDPM baseline, sampled objects get skewed and the movement is incorrect (matching videos are available on the website).

---

> > > > > > ### Comment · Reviewer_X2Za · 2023-11-22
> > > > > > **Unclear questions.**
> > > > > >
> > > > > > 1. These questions are still not answered. What's the target distribution? How to get it?
> > > > > >
> > > > > > 2. Authors should not assume that the ICLR reviewer and others should already know a SIGGRAPH23 paper published months ago with <10 citations till now. If the proposed method is extensively based on Heitz et al. (2023), then a self-contained present tation of that paper is required. The method section of the current manuscript only mentions this paper in lines around Theorem 4.1, and these lines are not self-contained. This sentence, " The only difference from Heitz et al. (2023) is the conditioning on C," looks so confusing. If the entire methodology only contains one difference on C, then it's too minor to be a conference paper in ICLR.
> > > > > >
> > > > > > 3. Authors should learn from Heitz et al. (2023). It's a good paper. All concepts are self-contained, and visualizations are quite direct and good, while experimental results are quite good.

---

> > > > > > > ### Author Response · Authors · 2023-11-22
> > > > > > >
> > > > > > > We address each point separately:
> > > > > > >
> > > > > > > 1. The target distribution is the distribution of the data.
> > > > > > >
> > > > > > > 2. Heitz et al. (2023) is mentioned extensively in the Introduction and the Background (section 2.2 is devoted to it).  Building on and using results from published work is a standard practice in scientific writing.
> > > > > > >
> > > > > > >
> > > > > > >     Our paper is self contained up to the proof of Thm 4.1. Since the proof of Thm 4.1 is a slight variation of a proof in Heitz et al.
> > > > > > >    (2023) [this is the sentence "The only difference from Heitz et al. (2023) is the conditioning on C," that the reviewer mentions], we
> > > > > > >    felt that a proof sketch is sufficient. However, we can easily add a full proof to the appendix.
> > > > > > >
> > > > > > >
> > > > > > >    This field moves fast. The other reviewers asked to **experimentally compare** with a paper from ICML 2023 (Consistency
> > > > > > >    Models). The Heitz et al. (2023) work is on arXiv since 2022, along with other similar work on deterministic diffusion (see
> > > > > > >    comment to reviewer 1n8o). Well versed readers should easily tackle the material in our paper.
> > > > > > >
> > > > > > > 3. Our contribution focuses on selecting the source distribution. For this investigation we devoted a whole section (Section 4) to illustrative experiments with graphics (and even videos in the supplementary material!). This was recognized, e.g., by reviewer HtfP. Our experimental results are state-of-the-art in diffusion policies.

---

> > > > > > > > ### Comment · Reviewer_X2Za · 2023-11-23
> > > > > > > > **Additional comments.**
> > > > > > > >
> > > > > > > > Thanks for your feedback and for helping me learn something in this domain.
> > > > > > > >
> > > > > > > > 1. I now know what the target distribution is. However, I still don't understand where the target distribution comes from in Eq. (1). I checked Heitz et al. (2023) paper, and there were no words such as "target" or "data distribution" in their paper. I believe that this is at least an unaddressed writing problem in this paper.
> > > > > > > >
> > > > > > > > 2. This paper needs an extensive rewrite, and authors should try to add their modules to other models, such as consistency models. A personal suggestion is not to present your major contribution as "improving the source distribution." This idea seems quite marginal, and I think this can hardly be accepted to a top-tier ML conference such as ICLR unless such an idea is proven across a wide range of methods/benchmarks or has a unique theory support. It's better to work on a seemingly more important idea in my eyes.
> > > > > > > >
> > > > > > > > 3. I agree with Reviewer 1n8o that this paper does not contain major novelty. I agree with Reviewer HtfP that this paper looks like a workshop paper. I agree with Reviewer eaWM that this proposed module needs to be applied to other methods to get it published.

---

### Official Review · Reviewer_eaWM · 2023-10-29

**Soundness:** 3 good
**Presentation:** 3 good
**Contribution:** 2 fair
**Rating:** 5
**Confidence:** 4

**Summary:**

This paper proposes a deterministic diffusion framework that samples (sequential) data more efficiently and effectively than established DDPM / DDIM frameworks.  The framework is based on IDAB, where mixed densities are iteratively deblended from the source to target distribution.   The core idea is to initialize the source distribution with those more closely resembling the target distribution, instead of starting from Gaussian noise.  Such a non-Gaussian distribution can be obtained using the available context or intermediate distributions sampled from a trained DDIM model.  This paper demonstrates better efficiency and stability of sequence predictions than DDIM/DDPM baselines.

**Strengths:**

1. The idea of denoising from a source distribution closer to the target makes sense.  Figure 1 and 3 clearly demonstrates the idea.
2. The paper is well written and easy to follow.

**Weaknesses:**

1. The compared baselines are incomplete.  Diffusion models have sparked great interest in the machine learning community, and there are multiple works focusing on improving the sampling efficiency of diffusion models.  For example, Progressive Distillation[1], DPM-Solver[2], 3-DEIS[3], and Consistency Models[4].  Especially, Consistency Model even requires only 1 diffusion step for sampling. I would expect Consistency Model to be the strong baseline across all experiments.

2. The efficacy of deterministic deblending module is unclear.  One can stack two separate DDIM modules by replacing the IADB with another DDIM.  Probably such a cascade (yet incremental) trick could also good performance and model efficiency.

3. It seems unclear to me when to using DDIM or sequence history to initialize the source distribution.  For video prediction, the authors use deblend from history.  For policy prediction, the authors use DDIM intermediate estimate.  There's no enough ablation study about why the authors choose to use DDIM intermediate estimate for action prediction.  I would expect it's because that using sequence history is more stable, but having troubles with capturing multi-modality.  On the other hand, using DDIM intermediate estimate seems to contradict with the abstract, where authors mentioned "Drawing on recent work on deterministic denoising diffusion, we initialize the denoising process with a non-Gaussian source distribution obtained using the context available when predicting sequence element".  DDIM starts denoising from Gaussian noise.


[1] Progressive distillation for fast sampling of diffusion models.  Salimans, Tim and Ho, Jonathan.  ICLR 2022.

[2] Dpm-solver: A fast ode solver for diffusion probabilistic model sampling in around 10 steps. Lu et al. NeuRIPS 2022.

[3] Fast sampling of diffusion models with exponential integrator. Zhang et al. ICLR 2023.

[4] Consistency Models. Song et al. ICML 2023.

**Questions:**

1. To clarify my concerns, I would expect the following experiements:

    * On BAIR and PHYRE, compare DDIM & DDPM & ASPeed that deblend previous history or Gaussian noise.  It would be clearer about the choice of deblending history (see Weakness 3).
    * On Robomimic and Push-T, compare to i) Consistency Model, and ii) two-stage DDIM, which resembles ASPeed but replaces the second-stage deterministic deblending module with another DDIM (see Weakness 1 and 2).  If stacked DDIM works, probably this paper should be positioned as cascade diffusion models instead.

2. One interesting (and I personally think would be more impactful) question is: can we generalize the first-stage distribution to some coarse-semantic / prior-knowledge distribution.  Take a simple example--face reconstruction, can we initialize the source distribution with mixtures of eigen faces (where we can introduce stochasticity)?  So that we can control the whole generation process (either start from some specific prior or randomly perturbed prior to start from pure noise).

---

> ### Author Response · Authors · 2023-11-19
>
> We thank the reviewer for their important feedback and their time. We address each point separately:
>
> Stacking DDIMs: assuming the suggestion is to run DDIM and use the result for initialization of another DDIM, we believe that this is not possible as DDIM expects a Gaussian source distribution and the output of the first DDIM would not be Gaussian. This is one of the reasons we use deterministic diffusion, which is a central component of our work.
> An alternative is to simply run DDIM for more steps - we compare with this baseline in Figure 8.
> If we misunderstood what the reviewer meant by stacked DDIMs, we kindly ask for clarification.
>
>
> BAIR and PHYRE: We thank the reviewer for this suggestion. We added the experiments on PHYRE in the supplemental rebuttal PDF (Section B.1). As mentioned in the general comment above, $DDIM_n$ Initialization in the video prediction domain is inferior to history-based initialization, and the initial approximation accuracy is important as there is a significant difference in the results of initializing with $DDIM_2$ or $DDIM_3$. We will add experiments on BAIR after the rebuttal as they take too long given our available compute resources.
>
> Robomimic and Push-T: We have added DPM-Solver++ and Consistency Models as baselines in the supplemental rebuttal PDF (Section C.2), which both perform worse compared to ASPeeD. We will update the relevant plots in the final version of the paper.
>
> Coarse Semantic/Prior Knowledge Distribution: Thank you for the interesting suggestion! We will definitely explore this direction in the future, it would be interesting to investigate “discrete” distributions as source distributions.

---

> ### Comment · Reviewer_eaWM · 2023-11-21
>
> Thanks for the response.  The followings are my response.
>
> ---
>
> **Other strong baselines**
>
> My concern is addressed
>
> ---
>
> **Stacking DDIM**
>
> Sorry for the confusion.  Since one variant of the method stacks two diffusion models with separate denoising scheduler, one question is that "Does the performance gain result from increased model capacity (two networks), or the novel combination of denoising schedulers?".  A direct ablation study is to compare with a stacking-DDIM baseline, where two networks are trained with separate DDIM denoising schedulers.  My guess is that if the hyper-paramters (e.g. the variance schedule, number of diffusion steps etc) are tuned optimally, stacking-DDIM might achieve comparable performance as the proposed method.
>
> ---
>
> **Source distribution**
>
> I think the ablation is somewhat disappointing, since history data is only available for sequential data.  For image generation, information retrieval would be a good solution [1].  I would encourage the authors to tackle this limitation and enrich the experiments.
>
> [1]: Re-Imagegen: Retrieval-augmented text-to-image generator.  Chen et al.

---

> > ### Author Response · Authors · 2023-11-22
> >
> > Thank you for taking the time to reply to our comments. We are glad that we were able to alleviate some of your concerns.
> >
> > Regarding stacking DDIM models: it is still unclear to us how such a stacking would be performed, as the output of a DDIM sampling procedure would not be Gaussian, and therefore, not an appropriate input for the second DDIM model. We suppose this type of stacking would be possible with some adaptation of the DDIM sampling method; this is an interesting avenue for future investigation, which is outside the scope of our work.
> >
> > Regarding the ablation on the video prediction datasets: as explained in the introduction, our work focuses on the sequence prediction setting, and our whole point is to show that history information can be exploited to improve generation quality/speed; whether directly or by conditioning the DDIM sampling to get a proper initialization point. In this context, the ablation strengthens our claim that the choice of initialization can be improved by using data available in the sequence. Investigating how our method can be applied to image generation is an interesting direction, but outside the scope of our work.

---

### Official Review · Reviewer_HtfP · 2023-10-31

**Soundness:** 2 fair
**Presentation:** 3 good
**Contribution:** 2 fair
**Rating:** 3
**Confidence:** 4

**Summary:**

This paper applies an iterative $\alpha$-(de)blending approach to generative modelling in sequential tasks (context conditioned video and action prediction). Here, the forward diffusion process is formulated as a linear interpolation between the source distribution in some context and a target distribution given the same context. The core approach is to speed up diffusion processes by drawing data from an improved source distribution, with two options considered. The first takes advantage of temporal similarity and uses a Gaussian noise perturbed prior sequence of states as the source distribution, while the second, more expensive approach, uses a pre-trained DDIM diffusion model. Results on video prediction tasks (BAIR and PHYRE) show improved video prediction quality with fewer diffusion steps, and improved rewards when compared to baseline models.

**Strengths:**

The core contributions of this work are to show that $\alpha$-(de)blending (Heitz et al. 23) is effective in the video domain, where stronger sequential priors over source distributions can be exploited. This is a sensible application of this approach, and the empirical evaluations in this work help to justify this.

The proposed approach produces good results, outperforming baselines.

Section 4.1 nicely explains the motivation behind the choice of improved distributions and how this reduces the number of steps required for generation.

**Weaknesses:**

The strength of the contribution is limited, and to my mind better suited to a workshop paper, particularly given that the proposed approach closely follows Heitz et al. 23.

The results are not particularly surprising, particularly when the relatively weak baselines used for comparison are considered. As mentioned in 5.1, the DDIM sampling approach starts from Gaussian noise, while the proposed approach bootstraps from the perturbed history. The core idea to bootstrap the diffusion process from a good initial guess has been applied to accelerate diffusion processes previously eg. Lyu et. al., which may be a better baseline to consider.

However, the idea of using another trained generative model (in this case DDIM) as a source distribution, while effective, seems extremely expensive, and needs to be justified/ better motivated. The comparisons in Fig 8 seem unfair, as the proposed approach takes 10 steps, but this requires an additional 3 steps for blending step (please correct me if I am wrong). This seems to indicate that 10 DDPM steps is doing comparatively well on this task.

The paper would benefit from significant levels of polish (proof reading, figure axis labelling, better choices of axis limits - eg. Fig 8a, these axes should reflect maximum and minimum rewards obtainable to put these results into context, figure order, making sure captions fully describe content).

**Questions:**

Fig 8. Why is 100 DDPM worse than 10 DDPM in push-T/ tool hang? How well does 30 DDIM/ DPPM (I think this is equivalent to 10 Aspeed steps?) steps do in comparison to Aspeed?

Fig 10. These ablations don't seem comprehensive enough to answer questions or support the claims. How dependent are these choices on the nature of the task?

---

> ### Author Response · Authors · 2023-11-19
>
> We thank the reviewer for their feedback and time.
>
> Novelty: See our general comment to all reviewers above.
>
> Baselines: Thank you for this suggestion. In the supplementary rebuttal PDF (Sections A and C.2) we address this concern by comparing our algorithm to Rectified Flow, DPM-Solver++ and Consistency Models, showing that our approach outperforms the latter two on robotic control tasks. As mentioned in the related work section, Lyu et al. [1] (as well as others) only focus on image generation, and have not attempted to extend their work to video prediction or robotic control. In particular, the method proposed by Lyu et al. requires pre-training a different (non-diffusion) generative model, which is avoided in our work by using the insight that deterministic diffusion does not require a Gaussian prior. To our understanding, the new baselines that we added are stronger than Lyu et al.’s ES-DDPM, but we can add an ES-DDPM baseline if the reviewer finds it pertinent.
>
> We emphasize that our approach can start with a distribution that is *not Gaussian*, which we exploit in the Diffusion Policy experiments. This is made possible by using deterministic diffusion. To our knowledge, this method of using deterministic diffusion to bootstrap from non-Gaussian source distributions is novel.
>
> Fig. 8: To clarify, ASPeeD takes a TOTAL of 10 steps. In the Push-T experiment, 2 DDIM steps and 8 deblending steps are taken, while in the Tool-Hang experiment 3 DDIM steps and 7 deblending steps are taken; in both cases the total is 10 which is a fair comparison with 10 DDPM or 10 DDIM steps. If this is unclear in the text, we can update the caption to better convey this point.
>
> Regarding 10 DDPM outperforming 100 DDPM: Inference with 10 steps of DDPM was not attempted by the authors of Diffusion Policy; we conducted this additional experiment for a full fair comparison. We found these results slightly surprising as well. Therefore we’ve re-verified the results during the rebuttal period, and found that 100 DDPM outperforms 10 DDPM on Tool Hang, and performs similarly to 10 DDPM on Push-T.
>
> Polishing: Thank you for your valuable suggestions. We will polish the writing accordingly and clarify the captions for the final version.
>
> Ablations: We added ablation studies for the Push-T task in the supplementary rebuttal PDF (Section C.1), in addition to the ablation study on the Tool-Hang task reported in the paper. The DDIM ablation on Push-T shows that the reward is not very sensitive to the ratio of DDIM-deblending steps, unlike the MSE. The perturbation ablation on the Push-T verifies the observation we made based on the Tool-Hang ablations – that there is no benefit in adding gaussian perturbation to the DDIM approximation.
> Additionally, we’ve added ablations on the PHYRE dataset with $DDIM_n$ initialization in the supplementary rebuttal PDF (Section B.1). $DDIM_n$ Initialization in the video prediction domain is inferior to history-based initialization, and the initial approximation accuracy is important as there is a significant difference in the results of initializing with $DDIM_2$ or $DDIM_3$.
>
> References:
>
> [1] Zhaoyang Lyu, Xudong Xu, Ceyuan Yang, Dahua Lin, and Bo Dai. Accelerating diffusion models via early stop of the diffusion process. arXiv preprint arXiv:2205.12524, 2022.

---

> > ### Comment · Reviewer_HtfP · 2023-11-22
> > **Thank you**
> >
> > Thank you for the clarifications in your rebuttal, and for the additional baselines. I think this is a good inclusion and strengthens the paper. I still have significant reservations about the importance of the contribution and paper clarity (eg. as pointed out by Reviewer eaWM - I do not think this paper can be understood with reading Heitz et al. as it stands)

---

### Official Review · Reviewer_1n8o · 2023-10-31

**Soundness:** 2 fair
**Presentation:** 3 good
**Contribution:** 2 fair
**Rating:** 5
**Confidence:** 4

**Summary:**

In this paper, the authors aim to expedite inference by leveraging the properties of the sequence prediction task.

**Strengths:**

1. The authors extend the iterative α-(de)blending approach, applying it to sequential tasks
2. The model shows improved NFE at inference time

**Weaknesses:**

1. While the authors' effort to extend a proposed diffusion model to a conditional or sequential generation task is commendable, it lacks a significant degree of innovation. The work does not present any new insights or methods, nor does it offer any architectural advancements that could potentially enhance the performance of conditional generation. The novelty factor of this study could be improved with the introduction of unique approaches or methodologies.

2. The iterative α-(de)blending method utilized in this research appears to bear a close resemblance to the rectified flow, as detailed in this [paper](https://arxiv.org/pdf/2209.03003.pdf). An important opportunity seems to have been missed by the authors in not acknowledging this significant work. A thorough review of related literature is crucial in any research endeavor to avoid overlooking key contributions in the field.

3. The concept of selecting an appropriate source or prior distribution is indeed valuable. However, it's worth noting that a similar discussion has already been conducted in [PriorGrad](https://openreview.net/pdf?id=_BNiN4IjC5). Furthermore, other methods from VAE research such as [SVG-LP](https://arxiv.org/pdf/1802.07687.pdf) provide more straightforward approaches to learning a source distribution. It is essential to acknowledge these existing methods and differentiate the new work from them.

4. The study's comparative analysis is somewhat lacking, with only DDPM and DDIM used as baseline samplers. The field has seen numerous new samplers designed to enhance sampling speed, such as rectified flow, [DPM-solver](https://arxiv.org/abs/2206.00927), and [Karras et al. 2022](https://arxiv.org/pdf/2206.00364.pdf). It is not necessary to compare the new model with all existing ones, but it would be beneficial to include a broader range of models in the comparison to provide a more comprehensive evaluation.

5. In terms of video prediction, the study could benefit from the inclusion of at least one higher-resolution dataset. A resolution of 128x128 would be sufficient to provide a more challenging and realistic evaluation of the model's performance. This would enable a more robust assessment of the model's capabilities in real-world applications.

**Questions:**

see above

---

> ### Author Response · Authors · 2023-11-19
>
> We thank the reviewer for their suggestions and valuable insights.
>
> Novelty: See our general comment to all reviewers above.
>
> Related Work:
> Rectified flow is indeed a beautiful paper that we were not aware of. Interestingly, it appears that an essentially very similar idea was investigated independently by several authors (Iterative α-(de)blending [1], Rectified Flow [2], Cold Diffusion [3] and Direct Iteration [4]), with insufficient cross-citations among them, unfortunately. Thank you for pointing out this work and others.  We duly extended our related work section; please refer to the rebuttal PDF (Section D).
>
>
> We emphasize, however, that our work is complementary to the line of work above - while all said papers investigate how to deterministically map different distributions, NONE OF THEM explore **the choice of the source distribution**, which is the main focus of our research. Our method builds on top of these papers while focusing on sequential settings. We demonstrate the fact that our method is independent of the blending algorithm by running our instructive example experiments (Section 4.2 of the paper) using Rectified Flow. See Section A of the supplementary rebuttal PDF for details.
>
> PriorGrad discusses the importance of the source distribution of the diffusion process in the audio prediction domain. In this work, the authors select the mean and variance of the Gaussian source distribution based on the data. Though they present a novel source distribution, it is still Gaussian and the algorithm is based on the standard diffusion process. In contrast, our approach is based on deterministic denoising models and can start the deblending process from non-Gaussian distributions, eg. $DDIM_n$ in the robotic control tasks, which achieves state-of-the-art results on this domain. Additionally, we focused on different domains - video prediction and robot control - for which it is not clear how to apply the PriorGrad source distribution.
> SVG-LP explored prior selection for video prediction based on VAE - it is not clear to us how this could be applied to our diffusion-based approach.
> Despite their dissimilarities from our approach, we’ve added both of these papers to the extended related work section; please refer to the supplementary rebuttal PDF (Section D).
>
> Baselines: Thank you again for pointing to additional related work. In the experiments described in the general comment (with results in Sections A and C.2 of the supplementary rebuttal PDF) we address this concern by comparing our algorithm to Rectified Flow, DPM-Solver++ and Consistency Models. Our method outperforms DPM-Solver++ and Consistency Models on the robotic control tasks.
>
> 128X128 Resolution video prediction results: We will assess how to extend our evaluation to these datasets - it is not trivial in terms of our available compute resources. Note that our results on the Diffusion Policy domains [5] are state-of-the-art, and evaluated on the most challenging tasks in the original paper.
>
> References:
>
> [1] Heitz, Eric, Laurent Belcour, and Thomas Chambon. "Iterative $\alpha $-(de) Blending: a Minimalist Deterministic Diffusion Model." arXiv preprint arXiv:2305.03486 (2023).
>
> [2] Liu, Xingchao, Chengyue Gong, and Qiang Liu. "Flow straight and fast: Learning to generate and transfer data with rectified flow." arXiv preprint arXiv:2209.03003 (2022).
>
> [3] Bansal, Arpit, et al. "Cold diffusion: Inverting arbitrary image transforms without noise." arXiv preprint arXiv:2208.09392 (2022).
>
> [4] Delbracio, Mauricio, and Peyman Milanfar. "Inversion by direct iteration: An alternative to denoising diffusion for image restoration." arXiv preprint arXiv:2303.11435 (2023).
>
> [5] Cheng Chi, Siyuan Feng, Yilun Du, Zhenjia Xu, Eric Cousineau, Benjamin Burchfiel, and Shuran Song. Diffusion policy: Visuomotor policy learning via action diffusion, 2023.

---

> > ### Comment · Reviewer_1n8o · 2023-11-22
> > **Thank you**
> >
> > Thank you for your response. While your answer has addressed some of my concerns, there remain significant issues that prevent me from endorsing the paper for acceptance. I have adjusted my recommendation to "weakly reject" with the following considerations in mind:
> >
> > 1. The paper would benefit from a more rigorous exploration of video prediction. Given that the authors assert the efficacy of their method in the context of video diffusion, it is imperative that additional experiments be conducted with more challenging datasets, such as the [Cityscapes dataset](https://www.cityscapes-dataset.com/). This would not only strengthen the current claims but also demonstrate the model's robustness and versatility.
> >
> > 2. The selection of the source distribution appears to have notable limitations:
> >     - The history-based approach is quite simplistic, relying on a perturbed version of the previous frame for initialization. This method overlooks the dynamics and trajectory inherent in the video history. The perturbation, derived from a standard Gaussian devoid of contextual relevance, could lead to inconsistencies in subsequent predictions. This issue might be exacerbated with more complex datasets. A potential improvement could be integrating techniques from models like SVG-LP, which learns a conditional prior in the context of the data. This learned prior may enhance the perturbation process.
> >     - The DDIM-based approach's reliance on an additional pretrained model raises questions about its suitability as a comparable method, especially when compared to other ODE/SDE solvers that do not require such dependencies.
> >
> > 3. The inclusion of non-diffusion methods as additional baselines is crucial. This is particularly pertinent for applications like robot control, where the inherent latency of diffusion models is still considerable when compared to one-step models like VAEs. The paper should provide compelling results that demonstrate the diffusion model's superiority or competitive performance in scenarios where latency is a less critical factor.

---

> > > ### Author Response · Authors · 2023-11-22
> > >
> > > We thank the reviewer for their suggestions, we’ll investigate them further.

---

### Author Response · Authors · 2023-11-19
**General Rebuttal Comment**

We thank the reviewers for their valuable insights and suggestions. We first address all reviewers and describe new results, following experiments we ran based on the reviewers’ suggestions. These are available in a supplementary PDF file (rebuttal_additions.pdf in the supplementary materials).

Novelty: This paper is the first to explore deterministic diffusion in **sequential tasks**: we present a principled framework based on existing ideas that are applied to this setting in a novel way with novel initializations for the source distribution. The finding that controlling the source distribution significantly improves performance in two important domains (video prediction and robotic control) is completely novel, and our paper presents state-of-the-art results for Diffusion Policies. For this domain, the best results we present utilize a non-Gaussian source distribution, made possible by our work (we may have under-emphasized this point in the text; we added specific results demonstrating this in the rebuttal PDF Section C.3).

Baselines: We’ve added a supplementary PDF with additional experiments and extended sections requested by the reviewers, which we will incorporate into the final version of the paper. These include:

1. [Section A] We provide a Rectified-Flow version of the instructive example (Section 4.2 of the paper), as requested by reviewer 1n8o.  The results show that our method is oblivious to the mapping algorithm between distributions - correct selection of the initial distribution is advantageous for improved results, **regardless of whether the underlying algorithm is Rectified Flow or Iterative $\alpha$-(de)blending**.  Corresponding videos can be found at [the project website](https://sites.google.com/view/aspeed-iclr2024).


2. [Section B] We added an ablation study on the PHYRE dataset, with $DDIM_n$ initialization, as requested by reviewer eaWM. In the PHYRE prediction task **we find no benefit in initializing the model from a $DDIM_n$ source**, as the results are inferior to the history-based initialization. As we hypothesize in the paper, this occurs due to the nature of the task: in video prediction frames are predicted one frame at a time, and the previous frame is a good approximation of the next frame.


3. [Section C] We added ablation experiments for the Push-T task as requested by reviewer HtfP, along with DPM-Solver++ [1] and Consistency Models [2] baselines for both robotic control domains, as suggested by reviewers X2Za, 1n8o and eaWM. The results show that our algorithm is superior to both order 2 and 3 of DPM-Solver++ with the multistep solver, and to Consistency Models trained with Consistency training (CT). **These results refute the concern that the advantages of our method could be achieved by stronger baselines.**

   Additionally, we include an ablation study of different initializations on the Push-T task as proposed by reviewers HtfP and eaWM. The results show that in Push-T $DDIM_n$ initialization outperforms history-based initialization as the rewards are higher and the MSE is lower, justifying our choice of the former for this domain.


4. [Section D] We extend the related work section following suggestions by reviewers 1n8o, HtfP and eaWM.


Finally, we plan to upload a revised version of the paper text itself as well, incorporating all of the above updates and changes suggested by the reviewers.

References:

[1] Cheng Lu, Yuhao Zhou, Fan Bao, Jianfei Chen, Chongxuan Li, and Jun Zhu. Dpm-solver++: Fast solver for guided sampling of diffusion probabilistic models. arXiv:2211.01095, 2022.

[2] Yang Song, Prafulla Dhariwal, Mark Chen, and Ilya Sutskever. Consistency models. ICML, 2023

---

> ### Author Response · Authors · 2023-11-22
> **Revision**
>
> We thank the reviewers for their time and feedback. We added a revision of the paper with the suggested modifications.

---

### Meta-Review · Area_Chair_X7oE · 2023-12-06

**Metareview:**

This paper presents an extension of the iterative α-(de)blending approach to sequential tasks, showing improved Normalized Flow Estimation (NFE) at inference time. Despite its strengths, such as the application to sequential tasks and improved NFE, the paper, with an average rating of 4.5, falls short of meeting the standards for acceptance at ICLR. The primary concerns revolve around the limited novelty of the contributions, the similarity to existing methods, and a lack of thorough comparative analysis with state-of-the-art techniques.

**Justification For Why Not Higher Score:**

The paper, while offering an extension of the iterative α-(de)blending approach to sequential tasks, does not demonstrate sufficient novelty or a substantial leap in methodology compared to existing works, which is necessary for a higher rating. The need for enhanced analysis with recent advancements and improvements in writing clarity and experimental depth are critical factors in determining the current score.

**Justification For Why Not Lower Score:**

N/A

---

### Decision · Program_Chairs · 2024-01-16

Reject